# Network models of protein phosphorylation, acetylation, and ubiquitination connect metabolic and cell signaling pathways in lung cancer

**Karen E. Ross**[1☯], **Guolin Zhang**[2☯¤a], **Cuneyt Akcora**[3], **Yu Lin**[1¤b], **Bin Fang**[4], **John Koomen**[5], **Eric B. Haura**[2], **Mark Grimes**[6]*

**1** Department of Biochemistry and Molecular & Cellular Biology, Georgetown University Medical Center, Washington, DC, United States of America, **2** Department of Thoracic Oncology, H. Lee Moffitt Cancer Center and Research Institute, Tampa, Florida, United States of America, **3** Department of Computer Science and Statistics, University of Manitoba, Winnipeg, Manitoba Canada, **4** Proteomics & Metabolomics Core, H. Lee Moffitt Cancer Center and Research Institute, Tampa, Florida, United States of America, **5** Molecular Oncology, H. Lee Moffitt Cancer Center and Research Institute, Tampa, Florida, United States of America, **6** Division of Biological Sciences, University of Montana, Missoula, Montana, United States of America

☯ These authors contributed equally to this work.
¤a Current address: mProbe Inc., Rockville, Maryland
¤b Current address: Epigenomics and Computational Biology Lab, Fralin Life Sciences Institute and Department of Biomedical Sciences and Pathobiology, Virginia-Maryland College of Veterinary Medicine, Virginia Tech, Blacksburg, Virginia
* Mark.Grimes@mso.umt.edu

**Data Availability Statement:** The data and code are publicly available on GitHub at https://github.com/m-grimes/PTMs-to-CCCN-and-CFN. Master

## Abstract

We analyzed large-scale post-translational modification (PTM) data to outline cell signaling pathways affected by tyrosine kinase inhibitors (TKIs) in ten lung cancer cell lines. Tyrosine phosphorylated, lysine ubiquitinated, and lysine acetylated proteins were concomitantly identified using sequential enrichment of post translational modification (SEPTM) proteomics. Machine learning was used to identify PTM clusters that represent functional modules that respond to TKIs. To model lung cancer signaling at the protein level, PTM clusters were used to create a co-cluster correlation network (CCCN) and select protein-protein interactions (PPIs) from a large network of curated PPIs to create a cluster-filtered network (CFN). Next, we constructed a Pathway Crosstalk Network (PCN) by connecting pathways from NCATS BioPlanet whose member proteins have PTMs that co-cluster. Interrogating the CCCN, CFN, and PCN individually and in combination yields insights into the response of lung cancer cells to TKIs. We highlight examples where cell signaling pathways involving EGFR and ALK exhibit crosstalk with BioPlanet pathways: Transmembrane transport of small molecules; and Glycolysis and gluconeogenesis. These data identify known and previously unappreciated connections between receptor tyrosine kinase (RTK) signal transduction and oncogenic metabolic reprogramming in lung cancer. Comparison to a CFN generated from a previous multi-PTM analysis of lung cancer cell lines reveals a common core of PPIs involving heat shock/chaperone proteins, metabolic enzymes, cytoskeletal components, and RNA-binding proteins. Elucidation of points of crosstalk among signaling

CFN/CCCN is on NDeX at https://www.ndexbio.org/viewer/networks/452ac60b-4681-11ed-b7d0-0ac135e8bacf.

**Funding:** M.G. and K.R. were supported by the NIH LINCS program U54 RFA-HG-14-001. M.G. is also supported by NIH R15DE028434 (with partial salary support) and a University of Montana Center for Translational Medicine Pilot Grant. K.R. is also supported by R35GM141873. G.Z. and EH were supported by Moffitt Innovative Core Project funding. Data for this work has been obtained with support in part by the Proteomics & Metabolomics Core Facility at the H. Lee Moffitt Cancer Center & Research Institute, an NCI designated Comprehensive Cancer Center (P30-CA076292, with salary support for J.K., B.F.). The funders had no role in study design, data collection and analysis, decision to publish, or preparation of the manuscript.

**Competing interests:** JK was funded in part by a sponsored research agreement with Bristol Myers Squibb unrelated to this work.

pathways employing different PTMs reveals new potential drug targets and candidates for synergistic attack through combination drug therapy.

## Author summary

Protein post-translational modifications (PTMs), such as phosphorylation, ubiquitination, and acetylation, are extensively employed by cell signaling pathways that regulate cell division, differentiation, migration, and cancer. We used machine learning to identify PTM clusters that represent functional modules in cell signaling pathways. These clusters were used to identify protein-protein interactions, and interactions between cell signaling pathways, that were active in lung cancer cells that were treated with anti-cancer drugs. We model these interactions as networks at three levels of granularity at the pathway, protein-protein interaction, and PTM levels. Interrogation of these networks yielded insights into molecular interactions between cell signaling pathways activated by oncogenes, transmembrane transport of small molecules, and glycolysis and gluconeogenesis. These analyses identify previously unappreciated mechanisms of crosstalk among signaling pathways between oncogenic tyrosine kinase signaling and proteins that regulate metabolic reprogramming in lung cancer, revealing new potential drug targets for combination therapy.

## Introduction

Protein post-translational modifications (PTMs) are intimately intertwined with cell signaling mechanisms that regulate cell differentiation, migration, and proliferation, and when mis-regulated become hallmarks and drivers of neurodegenerative diseases and cancer [1–8]. While the transcriptome provides a robust signature for different cell states or responses to perturbagens, gene expression is an indirect readout of signaling pathway activation, and the levels of mRNAs and proteins are not necessarily well correlated [9]. Analysis of PTMs is a more direct indicator of signaling pathway activation, since many signaling pathways are controlled by PTMs, and examining PTMs on a large scale reveals how cell signaling networks control diverse cellular responses [10]. While different types of PTMs (for example phosphorylation, acetylation, and ubiquitination) are traditionally studied independently, many proteins are modified by more than one type of PTM, and this information is integrated by cell signaling pathways. The ability to analyze different types of PTM information in the same samples adds richness to models of cell signaling pathways.

Lung cancer is one of the most commonly diagnosed cancers and the cancer responsible for the most deaths worldwide [11]. Non-small cell lung cancer (NSCLC), a sub-type of lung cancer, is frequently associated with activating mutations in receptor tyrosine kinases, including EGFR, MET, ALK, and ROS1 [12], which disrupt tyrosine kinase signaling and drive cancer progression. Targeted tyrosine kinase inhibitors (TKIs) have been developed that can be very effective treatments in the short term. However, cancers inevitably become resistant to these drugs either through further mutation of the target kinase or through activation of compensatory signaling pathways. To work towards the goal of development of new therapies to circumvent or overcome drug resistance, the motivation for this study is to acquire a better understanding of how cell signaling networks are modulated by mutated kinases and TKIs in lung cancer cell lines.

Our approach to examining cell signaling pathways with high resolution is to integrate PTM proteomic data, monitoring changes in multiple PTMs in response to TKIs, with protein

interaction networks [13–15] and biological pathways. PPI databases (STRING, GeneMania, BioPlex, and PathwayCommons) [16–20] attempt to define different types of relationships among proteins. PPI data are represented as a network where proteins are nodes and their interactions are edges. However, these networks represent the union of interactions observed in a wide range of cell types, disease states, and environmental conditions making it difficult to hone in on the signaling pathways that are most relevant to a particular experiment. We previously developed computational methods to address this issue. We took advantage of the fact that groups of interacting proteins that are coordinately post-translationally modified in the same samples identify signaling pathways activated in different tumors or cell lines and under different conditions [13–15]. By observing changes in multiple PTMs in lung cancer cell lines treated with TKIs, we can define clusters of coordinately-regulated PTMs, which reveal patterns specific to the drugs and cell lines under study. These clusters form a network that we refer to as a Co-cluster Correlation Network (CCCN). Proteins known to interact with each other whose PTMs co-cluster are likely to represent functional signaling pathways [15]. Clusters can also be used to construct a Cluster-Filtered Network (CFN) by filtering PPI edges from curated databases to include only edges between proteins whose PTMs co-clustered. This process simplifies the complex network of PPIs to focus on cell signaling interactions supported by PTM data.

We hypothesized that different TKIs would perturb cell signaling pathways, which would be reflected by PTM changes, to reveal mechanisms unique to the drug target and common core pathways that respond to all TKIs. To test this, lung cancer cell lines were treated with TKIs matched to inhibit the mutant oncogene. Phosphorylation, ubiquitination, and acetylation were chosen to examine crosstalk between signaling pathways involving these PTMs. We constructed a CCCN and CFN from measurements of phosphorylation, acetylation, and ubiquitination in 10 different lung cancer cell lines with driver mutations in EGFR, ALK, ERBB2/HER2, ROS1, and DDR2, treated with four different TKIs. In a new innovation, we used curated pathways from NCATS BioPlanet [21] to identify interactions among cell signaling pathways that are supported by our PTM data, which we call pathway crosstalk. This strategy provides a high-level overview of connections between specific pathways and cellular mechanisms whose details can then be explored using the CFN and CCCN models. This integrated model was interrogated with an eye towards interactions between pathways that do not have proteins in common but whose PTM patterns indicate a relationship. We also compared our current CFN with a CFN previously constructed using a different PTM dataset [15] and identified a core network that was common to both studies despite differences in PTM data acquisition methodology, PTM types observed, and cell lines and drugs used.

## Results

### Construction of the co-cluster correlation network (CCCN) and cluster filtered network (CFN)

Sequential enrichment of post translational modification (SEPTM) proteomics [22] was used to identify tyrosine phosphorylated, ubiquitinated, and acetylated proteins in lung cancer cell lines treated with four tyrosine kinase inhibitors (TKIs), crizotinib, erlotinib, dasatinib, and afatinib (Fig 1A and 1B). A total of 12,461 unique PTMs were identified (S1 Table). Of these, 4987 phosphorylation sites, 3249 acetylation sites, and 4452 ubiquitination sites were changed by at least 2.25-fold up or down compared to control cells by at least one TKI treatment (S1A Fig). Different PTMs responded asymmetrically to different TKIs; for example, dasatinib and afatinib predominantly inhibited phosphorylation and acetylation and concomitantly increased ubiquitination, while crizotinib and erlotinib caused changes in all PTMs in both directions (S1B Fig). There were 60 phosphorylation, 21 acetylation, and 13 ubiquitination

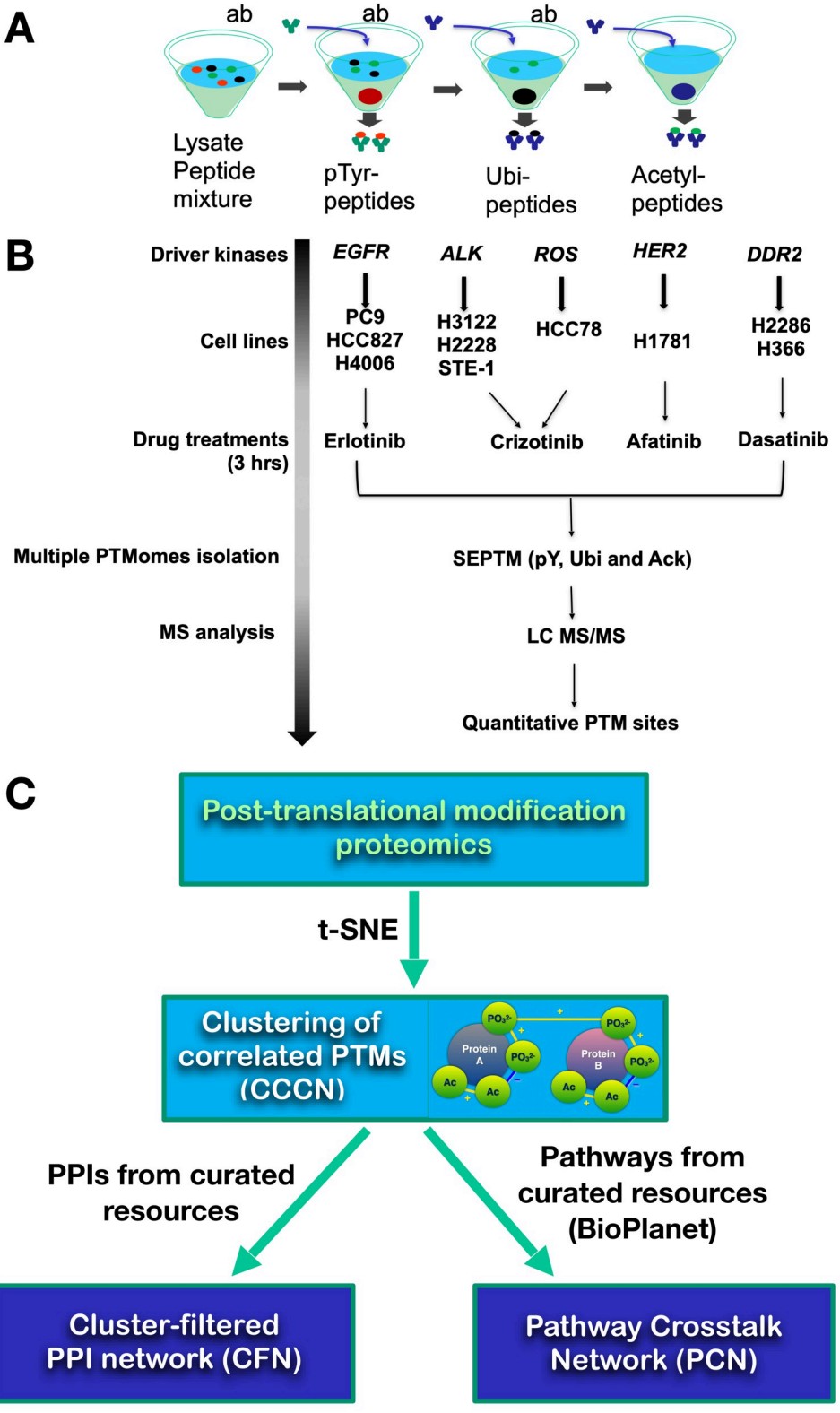

**Fig 1. Strategy.** Sequential enrichment of post translational modification proteomics (SEPTM-proteomics) procedure. (A) Principle of SEPTM peptide enrichment; modification-specific antibodies pull down specific PTMs sequentially. (B) Proteomics data generation procedure and drug-cell line combinations analyzed. (C) Graphical outline of

workflow. PTMs from PTM proteomics were subjected to t-SNE-based clustering to create the CCCN; a cluster-filtered network (CFN) was created by filtering PPI interactions to exclude all interactions save those from proteins with co-clustered PTMs. Similarly, relationships between BioPlanet pathways were defined by the extent to which their member proteins have PTMs that co-cluster in the CCCN. Namely, two pathways, A and B, have potential crosstalk if protein(s) from Pathway A and protein(s) from Pathway B have PTMs that co-cluster.

sites that were affected by all four TKIs (S1A Fig and S1 Table). Results of Gene Ontology (GO) enrichment analysis using Enrichr [23] on the genes associated with these multi-drug-affected PTMs suggest connections from tyrosine kinase-driven cell signaling to several other biological processes, including adhesion, gene expression, and cytoskeletal regulation. There were also three genes involved in glycolysis (ALDOA, GAPDH, GPI; S1 Table) that had PTMs affected by all drugs tested. We hypothesized that these data may be interrogated to find network connections between drug targets, downstream proteins, and cellular pathways. Thus, we analyzed these data using a combination of approaches that model PTMs, proteins, and pathways as networks (Fig 1C).

We used the dimension reduction algorithm t-distributed stochastic neighbor embedding (t-SNE) to help identify groups of PTMs whose abundances changed in a coordinated manner across different drug and cell line combinations (Fig 1C). For each PTM observed, we calculated the ratio of its abundance in each drug-treated sample to its abundance in the same cell line treated with DMSO (control). Ultimately, each PTM is described by up to 25 treatment vs. control ratios. t-SNE reduces the complexity of this high-dimensional data by embedding relationships among the PTMs in a three-dimensional graph [24,25]. As a result, PTMs that are near one another in the t-SNE embedding represent PTMs that share close statistical relationships (e.g., high Spearman correlation, low Euclidean distance) (S2A Fig). PTMs in the low dimensional embedding were clustered using the minimum spanning tree method. The full list of PTMs and their cluster membership can be found in S2 Table. The data were then modeled as a network (called a co-cluster correlation network, CCCN; Figs 1C and S2B) where PTMs (nodes) were joined by an edge if they belonged to the same cluster. S3 Fig shows all RTK and SRC-family kinase (SFK) PTMs, graphed as a heatmap (S3A Fig), and as PTM networks showing CCCN edges (yellow, positive correlation; blue, negative correlation) where node shape depicts protein family membership and node size and color indicate $\log_2$ fold change in response to TKIs (S2C and S3B and S3C Figs).

We then used the clusters to filter a network of physical protein-protein interactions obtained from the PPI databases STRING [19], GeneMANIA [17,26], BioPlex [18], and Pathway Commons [16], and kinase-substrate interactions from PhosphositePlus [27], only retaining edges between proteins whose PTMs co-clustered in the CCCN, to create a cluster filtered network (CFN; S4A Fig). The CFN and CCCN were combined (S4B Fig) to model the data in a similar manner as described previously for phosphorylation, acetylation, and methylation PTMs derived from lung cancer cell lines [15].

To test the self-consistency of the CFN, we compared the shortest paths subnetworks connecting the drug target (*i.e.*, the mutated cancer-driver protein) in each cell line to the proteins whose PTMs changed in response to drug treatment. Our results suggest that despite the inherent noisiness of the underlying PTM proteomic data, the CFN can reproducibly capture signaling responses to TKIs (S1 Text and S5 Fig).

## Pathway crosstalk

We next hypothesized that patterns of PTMs could be used to identify relationships among known biological pathways in human cells, which we call pathway crosstalk. Therefore, we extended our CCCN/CFN model of lung cancer PTM data to investigate relationships among

1657 curated pathways from NCATS BioPlanet, which consists of groups of genes (proteins) involved in various cellular processes [21]. To model interactions among pathways, we constructed a pathway crosstalk network (PCN) where pathway-pathway relationships (network edges) are defined by pathway genes whose PTMs co-cluster (Fig 1C). The edge weight on the interactions between pathways (called PTM cluster weight) is defined by the number of each pathway's proteins whose PTMs co-cluster (defined by t-SNE as described above), normalized to give less weight to genes found in many pathways and genes represented in large clusters (see Methods). The PCN consists of 645,709 pathway-pathway interactions with non-zero PTM cluster weight.

We compared the PTM cluster weights to two other measures of pathway-pathway similarity: i) Jaccard similarity, which captures the extent to which pathways have genes in common, and ii) Gene Ontology (GO) similarity, which measures the extent to which genes in the two pathways are annotated with common GO Biological Process terms (see Methods) [28]. These comparisons support the hypothesis that the PTM clusters return biologically relevant information that is complementary to, and independent of, common genes or common GO annotation, and that the PTM cluster weight will be useful for filtering relationships between pathways (S1 Text and S6 Fig).

To exemplify how the PCN can be used to discover novel signaling responses, we considered pathway interactions with the EGF/EGFR signaling pathway, which has a well-established role in lung cancer. In the PCN, the EGF/EGFR signaling pathway interacts with 1,363 other BioPlanet pathways with non-zero PTM cluster weight (S3 Table); of these, 620 have no genes in common with the EGF/EGFR signaling pathway (*i.e.*, Jaccard similarity is 0). Considering the interacting pathways with no genes in common that have the highest PTM cluster weights, a few themes emerge. There were pathways that mediate many major steps of gene expression (e.g., Transcription, Messenger RNA processing, Splicesome, Translation, and Cap-dependent translation initiation), pathways involved in glucose metabolism (e.g., Glycolysis and gluconeogenesis, Glucose metabolism, Gluconeogenesis, and Carbohydrate metabolism) and pathways that regulate transport of metabolic building blocks (e.g., Transmembrane transport of small molecules, SLC-mediated transmembrane transport, and Transport of inorganic cations/anions and amino acids/oligopeptides). We focused on two of these pathways—Glycolysis and gluconeogenesis and Transmembrane transport of small molecules (Fig 2A)—because of their strong PTM cluster-based links to the EGF/EGFR signaling pathway and because of the importance of metabolic adaptations in the progression to advanced cancer [29]. The PTM cluster weight of these pathways ranked fourth and ninth, respectively, among pathways that interact with the EGF/EGFR signaling pathway, but have no genes in common, and the PTM cluster weight of these pathway interactions with the EGF/EGFR signaling pathway were in the top 0.22% of PCN interactions among pathways with no genes in common (S7 Fig), and in the top 1% in the PCN overall. First neighbors of these pathways with high PTM cluster weight (purple edges) and/or a high number of genes in common (high Jaccard similarity; green edges) are shown in Fig 2B). As described below, interactions between the EGF/EGFR signaling, Transmembrane transport of small molecules, and Glycolysis and gluconeogenesis pathways were further explored using the CFN/CCCN data structure to determine the cell signaling pathways driven by PTMs that mediate pathway crosstalk.

## CFN/CCCNs between EGF/EGFR signaling and Transmembrane transport of small molecules

Examining all shortest paths in our combined CFN/CCCN model of data structure that connect proteins in the EGF/EGFR signaling pathway (top, Fig 3A) to those in Transmembrane

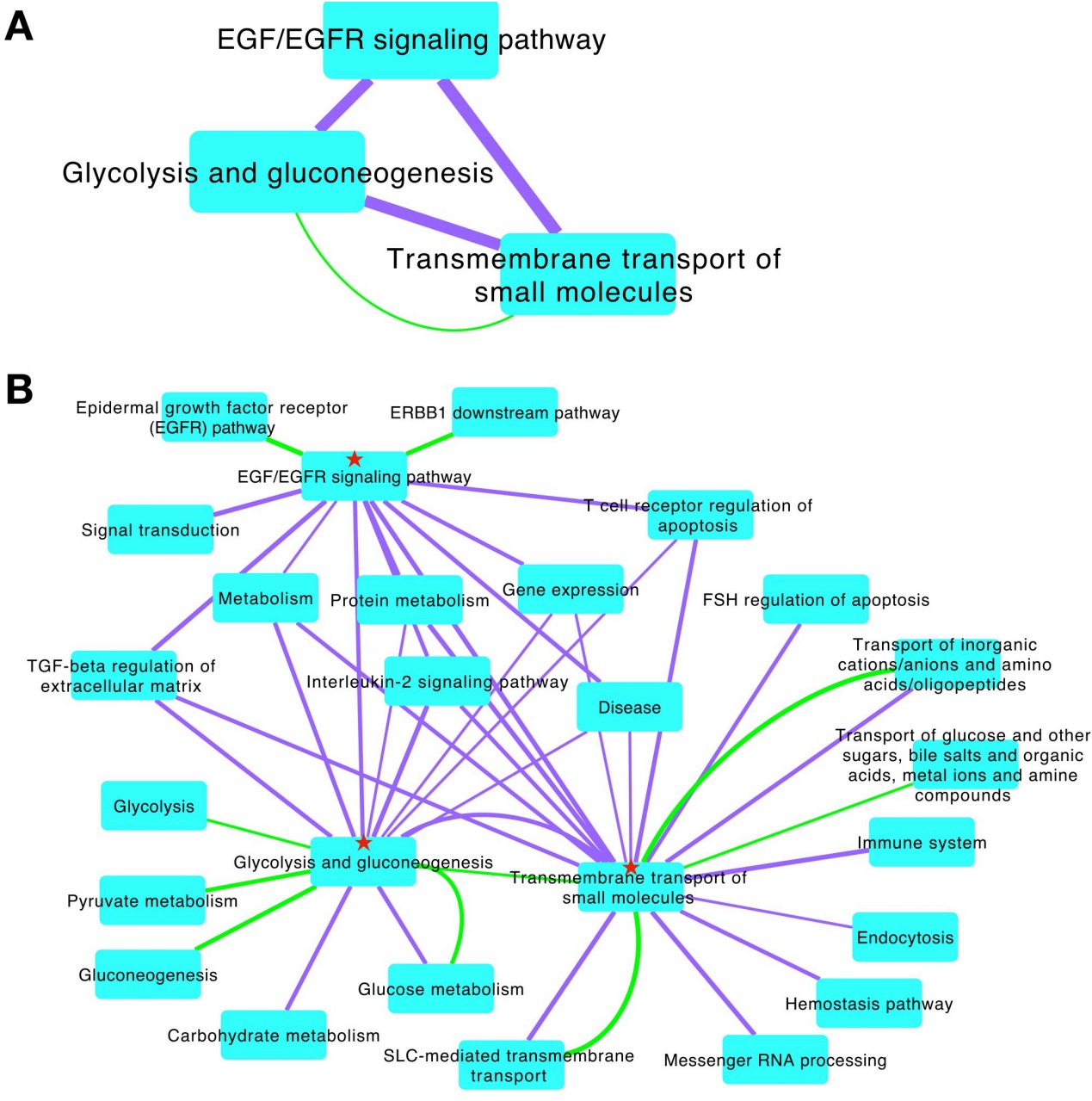

**Fig 2. Pathway crosstalk networks with the EGF/EGFR signaling pathway.** (A) Three pathways linked by strong PTM cluster weight (purple edges). Note that the EGF/EGFR signaling pathway has no genes in common with Glycolysis and gluconeogenesis and Transmembrane transport of small molecules, but the latter two pathways have 11 genes in common (green edges represent pathway Jaccard similarity). (B) Nearest neighbors of pathways in A (★) with additional edges filtered to show only strong associations (PTM cluster weight > 0.065 and/or pathway Jaccard similarity > 0.5).

transport of small molecules (bottom, Fig 3A) reveals striking complexity. Clusters of PTMs stand out as cliques with yellow positive correlation edges. As previously reported [15], we also noted reciprocally antagonistic relationships between different types of PTMs within the same protein (blue edges, Fig 3A). A number of PTMs stand out as being strongly inhibited by TKI treatment (large blue nodes Fig 3A and 3B), while a smaller number of PTMs increased (large yellow nodes, Fig 3 and S4 Table).

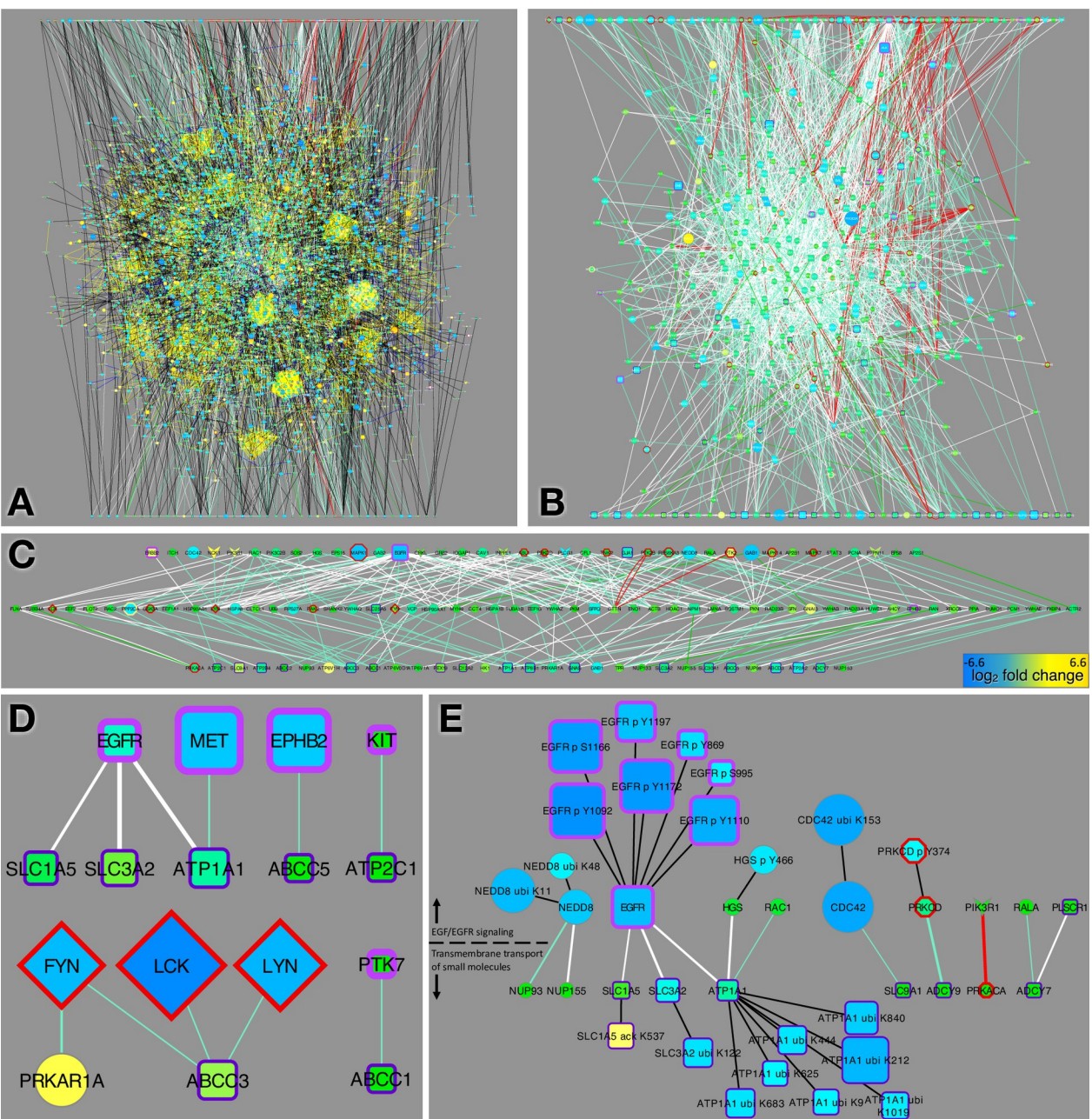

**Fig 3. EGFR signaling and small molecule transport pathway interactions.** (A) Combined CFN/CCCN showing composite shortest paths from the Bioplanet pathways EGF/EGFR signaling pathway (top row) and Transmembrane transport of small molecules (bottom row). PTM clusters are apparent as cliques connected by yellow correlation edges. (B) same as A but showing CFN edges only. (C) "Mutual friends" (center row) defined as proteins that connect to at least one member of each pathway in the CFN. (D) CFN interactions between RTKs, SFKs and the Transmembrane transport of small molecules pathway. (E) Direct CFN interactions between members of the EGF/EGFR signaling and Transmembrane transport of small molecules pathways. PTMs that were significantly changed (>2.25-fold) in response to erlotinib are shown. Node size and color represents the median log$_2$ fold change in response to erlotinib (A, B, C, E), or dasatinib (D); color scale bar shown in C. Node border and shape and edge colors are defined in S2C Fig.

To define cell signaling pathways connecting EGF/EGFR signaling pathway and Transmembrane transport of small molecules, we graphed the CFN of all shortest paths without PTMs (Fig 3B). From this network we identified intermediate nodes that directly connected to

one or more members of each pathway ("mutual friends"; Fig 3C). A number of cell signaling enzymes were in this group of mutual friends, including several SRC-family kinases (SFKs): FYN, LYN, and LCK (Fig 3C) and the RTK EPHB2. More broadly, there were numerous direct links in the CFN between RTKs and SFKs and the Transmembrane transport pathway proteins (Fig 3D), including connections between RTKs and five transporters (ATP1A1, SLC1A5, SLC3A2, ABCC5, ATP2C1) and interaction between SFKs and two others (ABCC1, ABCC3).

Several ABC transporters (ABCC1, ABCC2, ABCC3, ABCC5) implicated in TKI efflux and multidrug resistance [30] had links to EGFR through SFKs, GSK3A, and HSP90 proteins (S8A–S8C Fig). Interestingly, erlotinib decreased ubiquitination at distinct sites on ABCC1 and ABCC3 (S8A Fig), suggesting a role for erlotinib in stabilization of ABC transporters and promotion of chemoresistance. Similarly, crizotinib dramatically decreased ABC transporter ubiquitination, likely stabilizing ABCC1 and ABCC3 (S8B Fig). In contrast, no decreases in ABC transporter ubiquitination were observed in dasatinib-treated samples (S8C Fig).

We examined direct CFN connections between the two pathways (Fig 3E) focusing on the subset of direct CFN connections where both interaction partners had PTMs that were changed at least 2.25-fold by erlotinib. There were two such connections between EGFR and the transporters, ATP1A1 and SLC3A2 (Fig 3E). We observed reduced phosphorylation of several sites on EGFR, indicating that erlotinib inhibits EGFR activity, as expected, as well as decreased ubiquitination of the transporters. Interaction of EGFR with these transporters has been shown to be highest when EGFR is inactive [31]. Additionally, ubiquitination of ATP1A1 and SLC3A2 leads to their down-regulation and removal from the plasma membrane; thus, by reducing ubiquitination, TKI treatment may result in increased levels of active transporters in the membrane. In contrast, in H3122 (EML4-ALK fusion) cells treated with crizotinib, PTM changes on EGFR—increased phosphorylation of the activating site Y1016 and decreased phosphorylation of Y998, which is involved in EGFR endocytosis [32]—suggest that EGFR was activated (S9A Fig). Moreover, some ubiquitination sites on the ATP1A1 and SLC3A2 transporters were upregulated, suggesting that, in ALK-driven lung cancer cells, stabilization of these transporters may not play a significant role in the metabolic changes induced by TKI treatment.

## CFN/CCCNs between EGF/EGFR signaling and Glycolysis and gluconeogenesis

A similar approach was used to analyze pathway crosstalk between the EGF/EGFR signaling and Glycolysis and gluconeogenesis pathways. The composite shortest paths connecting these two pathways was also very complex (S10A Fig, CFN/CCCN and S10B Fig, CFN). The "mutual friends," nodes that connected to at least one member of both pathways in the CFN, contain some similarities and differences compared to those connecting the EGFR signaling and Transmembrane transport pathways (S10C Fig). Among the mutual friends that were in common were the SFKs FYN, LYN, and LCK and the RTK EPHB2.

Up-regulation of glycolysis in the presence of oxygen is a well-recognized phenomenon in cancer cells known as the Warburg effect [29]. The PTM changes we observed in proteins directly connecting the two pathways suggest that erlotinib treatment may paradoxically support the Warburg effect through crosstalk between EGFR signaling and glucose metabolism pathways (Fig 4A). Most of the PTM changes in the EGFR signaling proteins are consistent with reduced protein abundance or activity—e.g., reduced tyrosine phosphorylation of EGFR, increased ubiquitination of CFL1, and reduced phosphorylation of Y374 on PRKCD, which is likely to result in less recycling of EGFR from the endosome to the cell surface [33]. In contrast, most of the PTM changes in the glycolysis and gluconeogenesis pathway proteins are

indicative of increased protein abundance or activity. We noted decreased ubiquitination of ALDOA, GAPDH, ENO1, TPI1, and LDHB, suggesting that these proteins were stabilized by erlotinib treatment. This observation is consistent with other evidence showing that glycolysis is regulated by ubiquitination in cancer cells [29]. Moreover, acetylation of ALDOA, which inhibits enzyme activity [34], was reduced. Thus, direct inhibition of EGFR by erlotinib led to inhibition of members of the EGFR signaling pathway and a concomitant increase in activity of glucose metabolic enzymes in the interacting Glycolysis and gluconeogenesis pathway. These metabolic changes are consistent with promotion of the Warburg effect.

Although the specific proteins and sites affected differed, we saw a similar overall metabolic changes in (EML4-ALK fusion) cells treated with crizotinib (S9B Fig). TPI, GAPDH, and LDHB abundance is likely to be increased due to reduced ubiquitination, ALDOA activity likely to be increased due to reduced inhibitory acetylation on K147 [34] and ENO1 activity is likely to be increased due to increased acetylation [35].

Clues about the mechanism mediating the link between TKI treatment and unregulated glycolysis were found by examination of direct connections between RTKs, SFKs and the Glycolysis and gluconeogenesis pathway: the glycolysis enzyme PKM is a hub that integrates information from several RTKs and SFKs, and ENO1 and PGK1 are also key regulatory proteins (Fig 4B).

Recently, it was shown that accumulation of the glycolysis enzyme TPI1 in the nucleus is associated with poor prognosis and chemoresistance in lung cancer [36]. Our analysis suggests a mechanism for TPI1 relocalization upon erlotinib treatment. TPI1 interacts with the actin-modulatory protein coffin (CFL1; Fig 4A), which is required to translocate the enzyme to the plasma membrane, where it associates with the sodium-potassium ATPase [37]. Our PTM data indicates that CFL1 is ubiquitinated and down-regulated by erlotinib treatment, suggesting that localization of TPI1 to the plasma membrane may be reduced, freeing TPI1 to accumulate in the nucleus.

## Communication between EGFR and SFKs

In our analyses, we repeatedly observed that SFKs were important intermediaries in the communication between EGFR signaling and transport and metabolism. To better understand the role of SFKs, we constructed a subnetwork of the CFN showing paths between EGFR and proteins whose PTMs were most affected by the SFK inhibitor dasatinib (Fig 4C–4D). Fourteen of the proteins in this dasatanib-affected network, were also "mutual friends" directly linking EGF/EGFR signaling pathway to Transmembrane transporters of small molecules (Fig 3C), and seventeen were "mutual friends" linking to Glycolysis and gluconeogenesis (S9C Fig), reinforcing the notion that dastinib treatment could mediate pathway crosstalk between EGFR signaling and pathways that affect cell metabolism. Within this network, we observed several proteins (ERBB2/HER2, NCK1, IRS2) whose PTMs were elevated in erlotinib (Fig 4C) and depressed or unchanged in dasatinib (Fig 4D). Of particular interest, phosphorylation of the related EGFR-related RTK, ERBB2/HER2, on Y877 was increased by erlotinib, whereas this same site was strongly inhibited by dasatinib. Examination of the response of all RTKs (S3 Fig) revealed other RTKs whose phosphorylation is increased with erlotinib treatment and inhibited by dasatinib (AXL p Y481; EPHB5 p Y774), and several more RTK sites strongly inhibited by dasatinib (S3 Fig). Examination of the response of SFKs to TKIs revealed that erlotinib increased tyrosine phosphorylation on activating sites of FYN and LCK, whereas LYN and YES1 were phosphorylated on their C-terminal inhibitory sites (S3B Fig). In contrast, mainly LYN appeared to be activated in crizotinib-treated cells (S3C Fig).

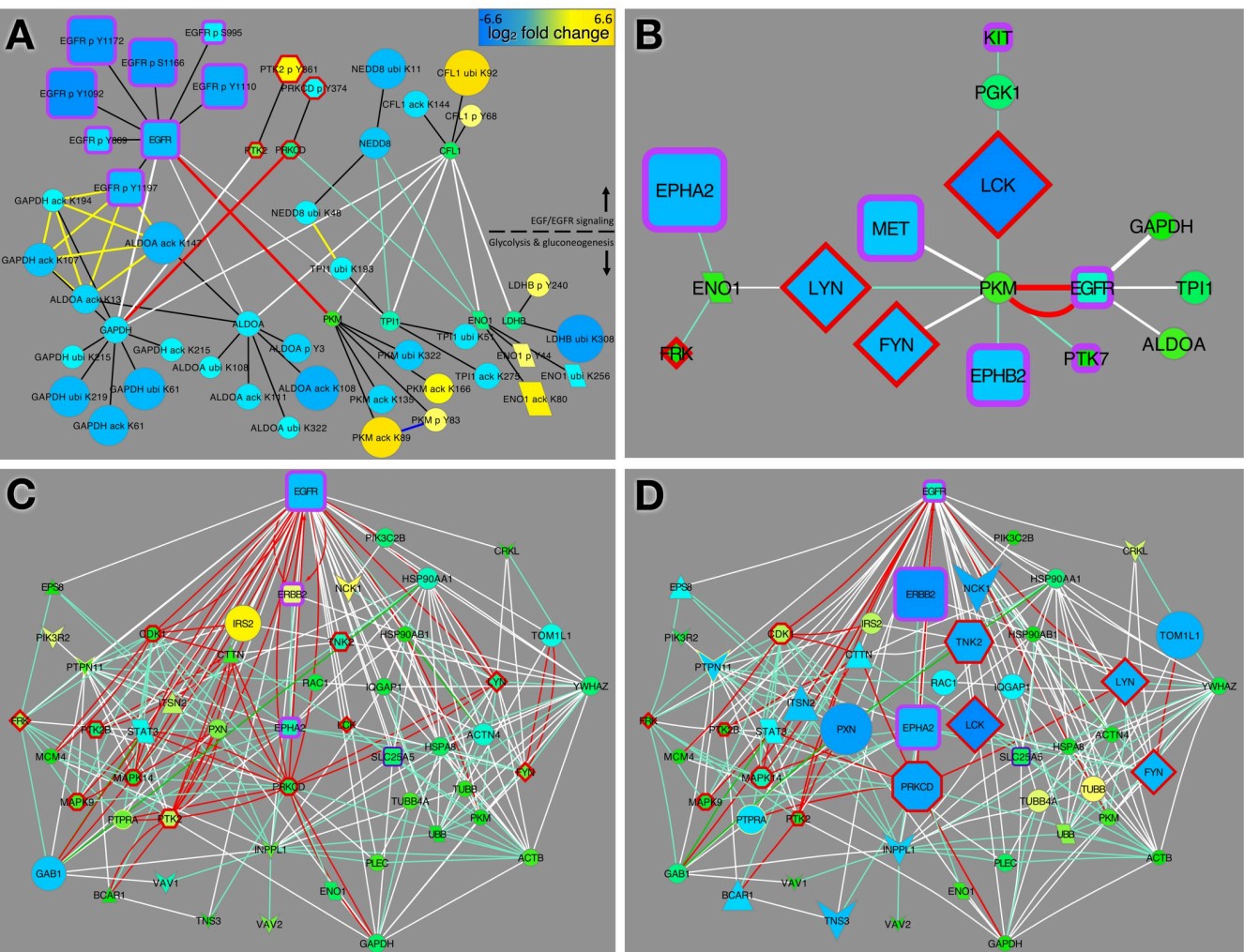

**Fig 4. EGFR signaling and glycolysis pathway interactions.** (A) Direct CFN connections between proteins in the EGFR signaling and glycolysis pathways and their PTMs that are significantly changed by erlotinib (median fold change in erlotinib-treated vs. control cells at least 2.25). (B) CFN connections between RTKs and SFKs and Glycoylsis and gluconeogenesis pathway proteins. (C, D) Sub-network from EGFR-glycolysis shortest paths CFN (see S10B Fig) showing links between EGFR and top proteins with dasatinib-inhibited PTMs. Node shape and edge color are defined in S2C Fig. Node size and color represents log2 fold change (bar in A) for all EGFR-mutant cell lines treated with erlotinib (A, C) or dasatinib (B, D).

## CCCN clusters enriched for drug-affected PTMs

We hypothesize that the PTM clusters in the CCCN represent signaling modules—groups of related proteins that respond to perturbations in a coordinated fashion. Thus, we investigated several clusters in depth in order to characterize the biological processes that they might be controlling. Because we were particularly interested in how signaling modules respond to TKIs, we focused on clusters that were statistically enriched for drug-affected sites (for details see S1 Text, S4 and S5 Tables). The three clusters of PTMs highlighted below illustrate different potential intersections among cell signaling pathways and biological processes.

The first cluster, which we call cluster A, was the most enriched cluster for PTM sites affected by crizotinib and dasatinib and was also enriched for erlotinib affected sites (Figs 5 and S11). The cluster contained activating phosphorylation sites on a number of known signaling proteins, including receptor tyrosine kinases (EGFR pY1197, EPHA2 p Y575),

downstream effector kinases (PTK2 pY576, TNK2 pY859, MAPK14 pY182, MAPK9 pY195), and tyrosine phosphatases (PTPN11 pY584, PTPN6 pY536, PTPN6 pY564, PTPRA pY798) (S6 Table).

The vast majority of the drug affected sites were down-regulated in the drug-treated vs. control cells (S11 Fig), which is consistent with the fact that TKI treatment inhibits signaling, reducing PTM of many sites in growth-factor signaling pathways. Notably, two of the few up-regulated sites were T14 and Y15 of the mitotic-driver kinase CDK1; phosphorylation of these sites inhibits kinase activity [38], so the effect of this PTM change was to suppress kinase activity. In addition to the signaling molecules themselves, Cluster A also contained direct or indirect substrates of several of the kinases that were targeted by TKIs in our experiments (S6 Table), e.g., substrates of ABL (Fig 5A, targeted by dasatinib), SRC-family kinases (Fig 5B, targeted by dasatinib), and EGFR (Fig 5C, targeted by erlotinib). The drug-affected sites were almost exclusively down-regulated by all TKIs examined, indicating that effects of suppressing one avenue of tyrosine kinase signaling propagated to the substrates of other tyrosine kinases. Importantly, some of the down-regulated substrates were implicated in negative regulation of growth-factor signaling (Fig 5D). For example, an activating phosphorylation site on TNK2, Y859, was down-regulated in all sample groups. TNK2 localizes to clathrin-coated pits and plays a role in internalization of growth factor receptors [39], so reduction in its activity may increase the amount of RTKs on the cell surface and prolong signaling. Suppression of these proteins that inhibit signaling may contribute to acquisition of drug resistance. In summary, Cluster A represents a group of signaling mediators and regulators whose activity is suppressed by treatment with multiple TKIs.

Cluster B was remarkable because of the large number of sites that were up-regulated by erlotinib treatment (S12 Fig). Eighteen out of the 44 sites in this cluster were up-regulated by more than 2.25-fold in the all.erl sample group, which included erlotinib treated PC9, HCC4006, and HCC827 cells. Many of the same sites were also up-regulated to a lesser extent in the pc9.erl (erlotinib-treated PC9 cells) sample group. Although Cluster B was also enriched for sites affected by crizotinib and dasatinib, these sites were largely down-regulated as was much more typical in our dataset. We performed pathway and GO enrichment analysis on the proteins with erlotinib-up-regulated PTMs using Enrichr [23]. The most enriched Bioplanet pathway was the ERBB1 (EGFR) downstream pathway (adjusted p-value = 0.014). Thus, many of these proteins are part of the EGFR pathway but have PTMs that are increased when EGFR is inhibited. The great majority (14/18) of these PTMs were tyrosine phosphorylations. Interestingly, when we examined sites from the EGF/EGFR signaling pathway and Transmembrane transport of small molecules pathway that co-clustered in CCCN (S8D–S8F Fig), we also observed tyrosine phosphorylation sites specifically up-regulated by erlotinib treatment but not other TKIs, namely INPPL1 (SHIP-2), the phosphoinositide (PI)-5 phosphatase that acts on PI(3,4,5) triphosphate, and two transporters: SLC20A2 (sodium-phosphate symporter) and SLC4A7 (sodium- and bicarbonate-dependent cotransporter; S8D Fig). Although the precise roles of these PTMs has not been characterized, increased tyrosine phosphorylation is often associated with signaling pathway activity; thus, these PTMs could be sentinels of signaling crosstalk that promote erlotinib resistance.

One of the top enriched GO Cellular Component terms for the set of proteins with erlotinib up-regulated PTMs was focal adhesion (adjusted p-value = 0.042). Focal adhesions serve as an intermediary between the actin cytoskeleton and the extracellular matrix and play a key role in migration and metastasis of cancer cells [40]. In total, the cluster contained 12 proteins with drug-affected PTMs that were associated with focal adhesions and/or actin cytoskeleton dynamics according to their UniProt function description and/or their GO term annotation (Fig 6A and S6 Table). Thus, Cluster B could be a module involved in cell motility and

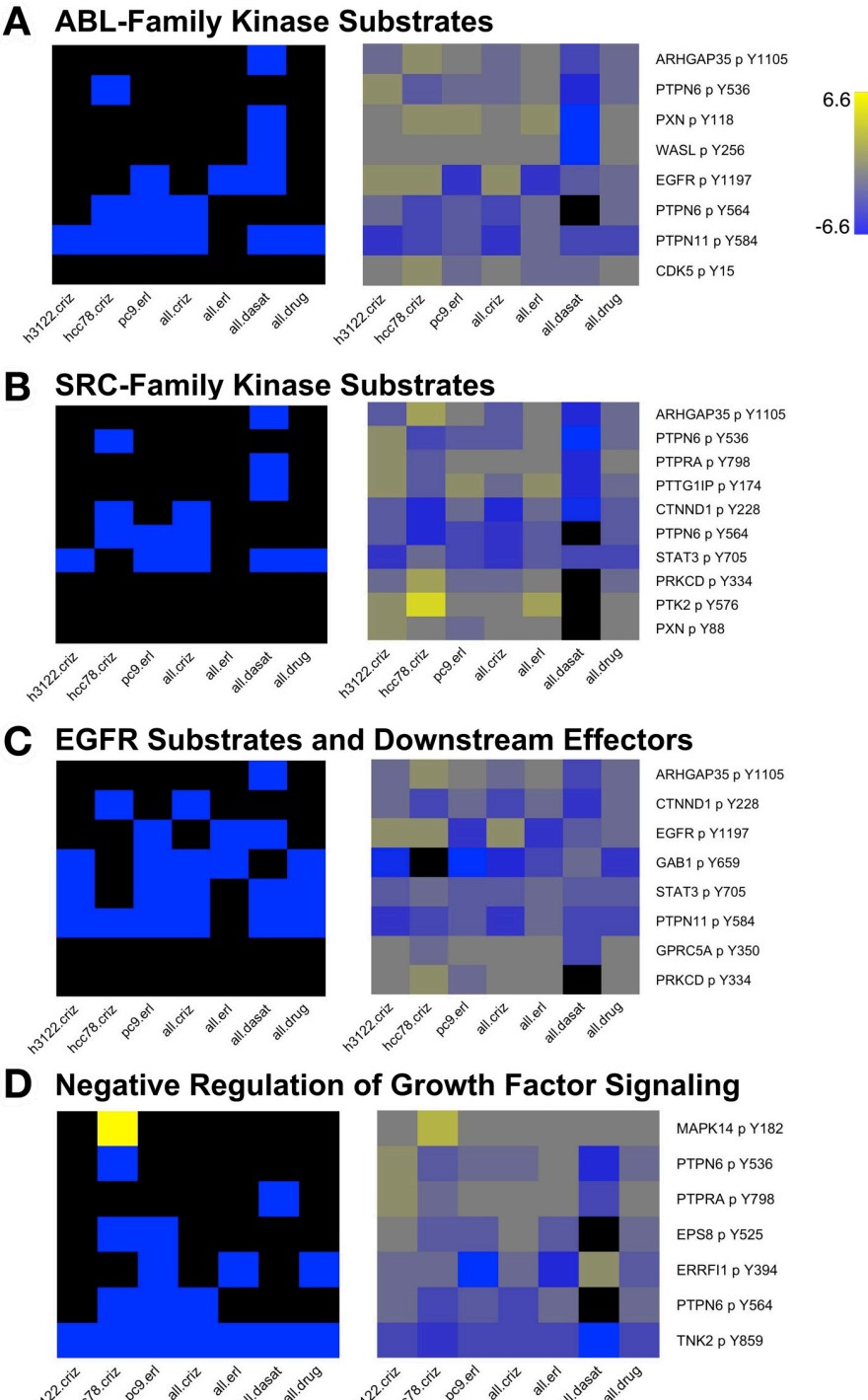

**Fig 5. Heatmaps showing subsets of drug-affected phosphorylation sites in cluster A, grouped by cell signaling pathway.** In the left hand panels, sites in cluster A whose median abundance ratio in drug-treated vs. control cells was at least 2.25-fold lower are blue and those that were at least 2.25-fold higher are yellow (black is below threshold). The right hand panels show the drug-treated to control ratio (median of samples in each group) for each PTM site (see key at right; black represents missing data). (A) Sites known to be phosphorylated by ABL-family kinases. (B) Sites known to be phosphorylated by SRC-family kinases (SFKs). (C) Sites that are phosphorylated directly by EGFR or downstream of EGFR activation. (D) Sites involved in negative regulation of growth factor signaling. Sample groups: h3122.criz: H3122 cells treated with crizotinb; hcc78.criz: HCC78 cells treated with crizotinib; pc9.erl: PC9 cells treated with erlotinib; all.criz: H3122, HCC78, H2228, and STE1 cells treated with crizotinib; all.erl: PC9, HCC4006, and

HCC827 cells treated with erlotinib; all.dasat: H2286 and H366 cells treated with dasatinib; all.drug: all samples from all.criz, all.erl, and all.dasat as well as H1781 cells treated with afatinib.

communication with the extracellular matrix. TKI-induced changes in the PTMs in this cluster could alter a tumor's potential for metastasis.

Finally, cluster C had a very different PTM composition, consisting of 29 sites the overwhelming majority of which (twenty-five) were acetylation sites (Figs S13 and 6B). The remaining four sites in the cluster were ubiquitination sites. Many of the proteins represented in this cluster have a role in transcription regulation (S6 Table). Half of the acetylation sites (12/29) were on histone subunits (H1, H2A, H2B, and H4). In cases where the role of the acetylation has been characterized (e.g., H2AFV K5/K8/K12, H2AFZ K5/K8/K12, and HIST4H4 K6/K9), the modifications are associated with open chromatin and transcriptional activation [41]. The cluster also contained two histone acetyltransferases (HATs): KAT7 and EP300, which have multiple interactions with histones (S13 Fig). The modification site on EP300, K1551, is in a region of the protein that is known to undergo auto-acetylation, increasing EP300 activity [42]. Thus, this cluster captures the relationship between changes in HAT activity and changes in levels of histone acetylation. The sites in this cluster were markedly downregulated in H3122 cells treated with crizotinib and PC9 cells treated with erlotinib (Fig 6B). Given the known roles of the proteins and PTMs in this cluster, this outcome would suggest that treatment with both drugs promoted an overall downregulation of transcription.

## Comparison of networks

We previously used similar methods to create a CFN from a dataset of phosphorylation, acetylation, and methylation PTMs in lung cancer cell lines (referred to here as the CST-CFN) [15].

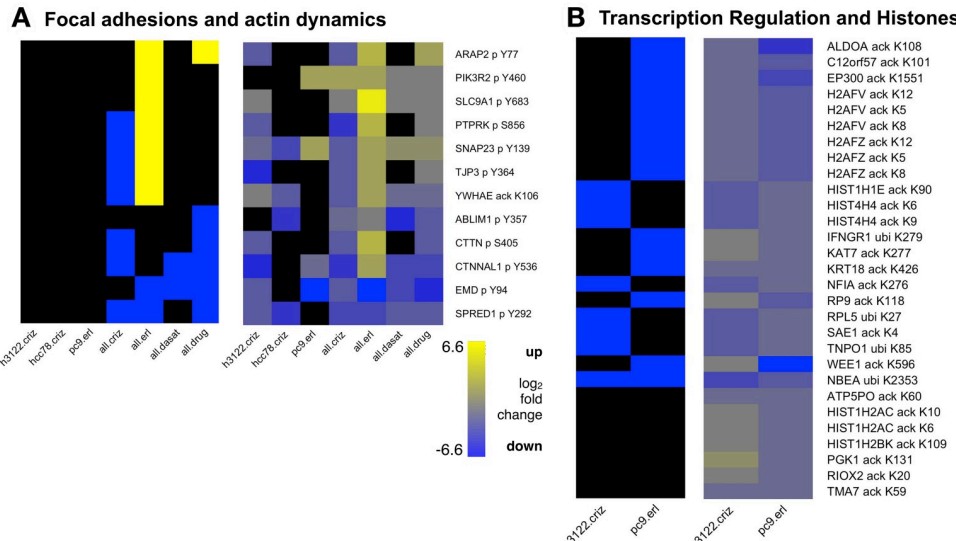

**Fig 6. Heatmaps showing PTM sites in Clusters B and C.** In the left hand panels, sites whose median abundance ratio in drug-treated vs. control cells was at least 2.25-fold lower are blue and those that were at least 2.25-fold higher are yellow (black is below threshold). The right hand panels show the drug-treated to control ratio (median of samples in each group) for each PTM site (see key at bottom; black represents missing data). Sample groups are as in Fig 5. (A) Sites in cluster B changed at least 2.25-fold in at least one sample group that are found in focal adhesions and/or are involved in actin dynamics. (B) Heatmaps of all PTM sites in Cluster C. Many of these sites are involved in regulation of transcription and/or are histone subunits. Sample groups: h3122.criz: H3122 cells treated with crizotinb; pc9.erl: PC9 cells treated with erlotinib.

(For a comparison of the basic statistical properties of the two networks, see S1 Text). While the data described here differs from that work in mass spectrometry methods and types of PTMs analyzed (methylation in the CST-CFN vs. ubiquitination here), both data sets contain phosphorylation and acetylation sites from lung cancer cell lines, so we hypothesized that we could identify a core set of the most highly connected nodes common to both networks. We employed k-core decomposition, which ranks nodes of a graph according to their connections and creates groups of increasingly more "important" or "central" nodes [43]. k-core decomposition has emerged as a fundamental operation in many areas such as graph similarity matching [44], network visualization [45], graph clustering [46], anomaly detection [47], and robustness analysis [48].

We applied k-core decomposition to the CFN generated from data in this study and the CST-CFN to identify the compositional differences between the two graphs in terms of the highest cores (see Methods, S1 Text, S14A and S14B Fig). The 43 genes in common to the high cores of both networks (cores 11–12 of our network and cores 7–10 of the CST-CFN) were highly connected to EGFR in our CFN (Fig 7). The common core contained several heat shock proteins (CCT4, HSP90AA1, HSP90AB1, HSPA8, HSPD1, NPM1), which is consistent with their role as chaperone proteins that bind diverse substrates in the cytosol. Consistent with the interplay between metabolism and cell signaling discussed above, the common core also contained the glycolysis proteins ENO1, FASN, PKM, and GAPDH. The core also contained six genes whose PTMs changed in response to all TKIs (HNRNPA2B1, FASN, GAPDH, HNRNPU, PPIA, ACTB, S1 Table) as well as ribosomal proteins, splicing factors and other RNA-binding proteins, cytoskeletal components including clathrin heavy chain and lamin A, and two 14-3-3 phosphoserine-binding proteins (Fig 7). Most of these genes had many PTMs that responded to more than one TKI and were positively-correlated with one or more PTMs of EGFR (S14C Fig).

Notably, there were a large number of negative correlations among different PTM types on these core proteins (S14C Fig, blue edges). Negative correlations between phosphorylation, acetylation and ubiquitination on the same protein indicate that these different PTMs were rarely found on the same protein in the same samples (S1C Fig). Thirty-nine proteins whose PTMs were negatively correlated were modified by both acetylation and ubiquitination on the same amino acid (S1 Table). Remarkably, ten of these were in the core (PKM, ACTB, DDX5, EEF1A1, HNRNPA1, HNRNPU, HSPA8, MYH9, SET, YWHAZ; S14C Fig). This supports the hypothesis that reciprocally antagonistic pathways control cell signaling through proteins that act as hubs or pathway control switches for signal integration [15]. Proteins with multiple PTM types will have different sets of interacting partners that depend on PTM-specific binding domains (e.g., ubiquitin binding domains; bromodomains; SH2 and PTB domains) to regulate protein localization, degradation, and activity.

## Discussion

This integrated analysis of protein phosphorylation, acetylation, and ubiquitination in lung cancer cell lines in response to TKIs attempts to model cell signaling pathways using network-based approaches. The activity of cell signaling pathways can only be indirectly inferred by interrogating the transcriptome or the proteome. While analysis of PTMs does not comprehensively cover all signaling pathways, or directly assess the role of intracellular second messengers, the integrated analysis of several PTMs enables modeling cell signaling pathways that are likely to be upstream of transcription and translation with sufficient resolution to begin to describe the interactions among pathways that affect many cellular processes.

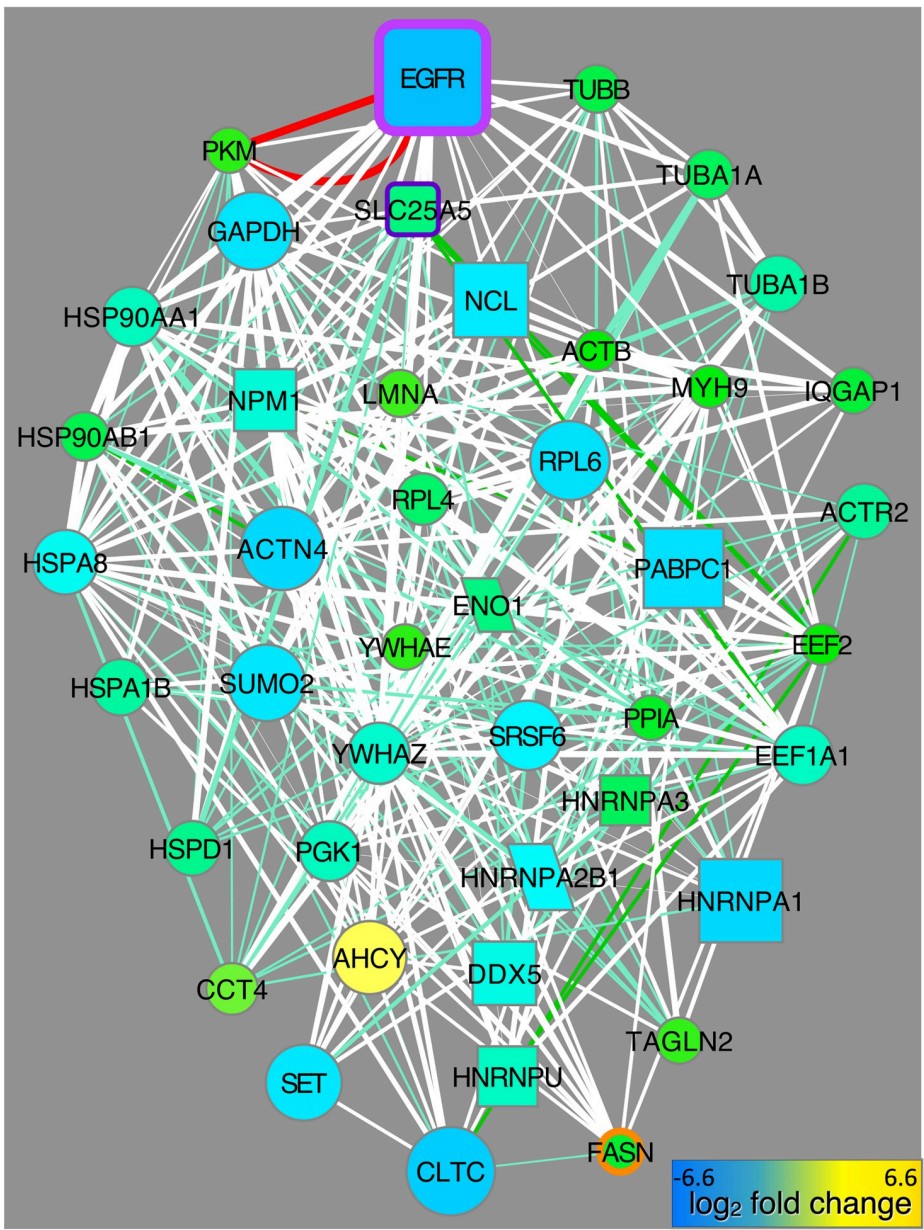

**Fig 7. Common core CFNs from different lung cancer PTM data sets.** The 43 proteins (excluding the ribosomal protein clique) that were in the highest cores of both CFNs graphed with CFN edges from this study. Node size and color indicates PTM changes in PC9 cells treated with erlotinib. Node border and shape and edge colors are defined in S2C Fig.

Our network-based framework allows interrogation of multi-PTM proteomics data with three levels of granularity at the pathway (PCN), protein (CFN), and PTM (CCCN) levels. The highest-level network, introduced for the first time in this study, is the PCN, in which the highly-curated biological pathways from BioPlanet [21] were connected based on the extent to which their member proteins' PTMs co-cluster in the CCCN. We focused on pathways that did not share any genes in common but nonetheless were strongly connected by PTM cluster

evidence, thus elucidating new interactions between pathways. A PPI-based view is provided by the CFN, in which PPIs from curated databases were filtered to retain only interactions between proteins whose PTMs co-cluster. This slimmed-down version of the full interactome selects interactions most relevant to the cell signaling response to TKIs in our experiments. We have previously shown that the CFN derived from PTM clusters effectively removes biases in PPI networks where well-studied proteins have more interactions [15]. The network that is the most detailed and the closest to the raw data is the CCCN, in which PTMs were clustered by t-SNE based on statistical relationships in TKI-treated lung cancer cells. Meaningful clustering of PTMs is challenging due to the sparseness of data sets typically obtained from mass spectrometry and immunoprecipitation using modification-specific antibodies. We have extensively evaluated this problem in previous work [13–15]. Here, we observed that drug-affected PTMs were asymmetrically distributed in the CCCN (*i.e.*, some clusters were highly enriched for drug affected sites), supporting the hypothesis that the clusters define functional modules whose members respond in concert to drug treatment.

Our integrated analysis of the PCN, CFN, and CCCN led to several biological insights that point to mechanisms of drug resistance and suggest possibilities for combination therapy. First, TKI treatment of lung cancer cells induces metabolic changes associated with advanced cancer and chemotherapy resistance, including up-regulation and re-localization of small molecule transporters and glycolysis enzymes. Although the details differed, up-regulation of glycolysis (*i.e.*, the Warburg effect) was apparent in both EGFR-mutant cells treated with erlotinib and EML4-ALK fusion cells treated with crizotinib. Our data revealed detailed mechanisms mediated by coordinated changes in phosphorylation (e.g., decreased EGFR activity), acetylation (e.g., reduced inhibitory acetylation of ALDOA and increased activating acetylation of ENO1), and ubiquitination (e.g., reduced ubiquitination of the glycolysis enzymes ALDOA, GAPDH, ENO1, TPI1, and LDHB and the transporters ATP1A1 and SLC3A2) (Figs 3, 4, S8–S10). Another potentially relevant interaction in our network is the kinase-substrate relationship between protein kinase C-delta (PRKCD) and GAPDH (Fig 4A). We observed reduced tyrosine phosphorylation of PRKCD, which could result in reduced phosphorylation of GAPDH. Other studies have shown that phosphorylation of GAPDH by PRKCD inhibits the turnover (mitophagy) of mitochondria [49]; thus, erlotinib-induced reduction in phosphorylation of GAPDH may lead to up regulation of mitophagy, which is a feature of cancer cells that are undergoing metabolic stress from their nutrient poor environment and/or the effects of anti-cancer treatments [50].

We also observed potential erlotinib-induced effects on the localization of transporters and glycolysis enzymes that promote tumor growth. First, erlotinib treatment may result in increased localization of transporters ATP1A1 and SLC3A2 to the plasma membrane (Fig 3E). Both ATP1A1 and SLC3A2 have been implicated in lung cancer. Many lung cancers show increased expression of ATP1A1, and targeting ATP1A1 with siRNA or drugs (cardenolides) has an anti-tumor effect [51]. SLC3A2 heterodimerizes with SLC7A5/LAT1, which has a well-established role in lung cancer [52–54]. Conversely, our results suggest that erlotinib treatment may result in localization of glycolysis enzymes away from the plasma membrane (Fig 4A). While increased aerobic glycolysis is a characteristic of cancer cells (Warburg effect), some glycolysis takes place even in healthy cells, particularly at the plasma membrane, where several glycolysis enzymes have been found associated with pumps and transporters that have high demands for ATP [37,55]. Altered localization of glycolysis enzymes could be an important factor in the metabolic reprogramming of cancer cells. It would be interesting to test whether erlotinib does indeed alter the localization of glycolysis enzymes, and whether that contributes to the reduced ubiquitination of these enzymes that we observed, which could increase aerobic glycolysis.

Our results point to the importance of SFKs as key signaling intermediates in TKI-treated cells. First, multiple SFKs were "mutual friends" with direct connections to members of the EGFR signaling pathway and both the Transmembrane transport of small molecules pathway (Fig 3C) and Glycolysis and gluconeogenesis pathway (S10C Fig). Second, SFKs were found on the shortest paths connecting EGFR to the ABCC transporters (S8 Fig). Third, SFKs are directly connected in the CFN to several transporters (Fig 3D) and glycolysis enzymes (Fig 4). Fourth, several RTKs show increased phosphorylation in erlotinib, but decreased phosphorylation in dasatinib (Figs 4B–4D and S3), suggesting that SFKs might be responsible for RTK activation in erlotinib-treated cells. Finally, SFKs show a reciprocal activation pattern in erlotinib vs. crizotinib, suggesting that FYN and LCK are responsible for increased tyrosine phosphorylation of RTKs in erlotinib-treated cells, whereas LYN phosphorylates different RTK sites in crizotinib-treated cells (S3B–S3D Fig). This is consistent with previous work showing that FYN and LYN show different responses in activity, association with the scaffold protein PAG1, and intracellular localization when activated by different RTKs in neuroblastoma cells [14,56]. It is also consistent with work showing that SFKs remain active in the presence of EGFR TKIs and contribute to EGFR-independent compensatory mitogenic signaling [57].

We also explored how several of the PTM clusters in the CCCN defined functional modules that respond to drug treatment (Figs 5, 6 and S11–S13). One cluster (cluster A) appeared to be a key signaling module. It contained many PTMs implicated in RTK signaling as well as numerous SFK target sites. Most of the PTM changes upon TKI treatment are consistent with down-regulation of signaling; however, some of the proteins with down-regulated PTMs are involved in feedback inhibition of signaling (e.g., TNK2 Y859) suggesting a mechanism for acquired TKI resistance. A second cluster (cluster B) contained a number of PTMs on proteins associated with focal adhesions and remodeling of the actin cytoskeleton. Many of these PTMs were up-regulated by erlotinib treatment and may provide a clue to the mechanism by which the epithelial-mesenchymal transition (EMT) facilitates acquired EGFR-TKI resistance in NSCLC [58]. Because they regulate cell shape and motility and mediate communication between the cytoskeleton and the extracellular matrix, focal adhesions play an important role in invasion and metastasis of cancer [59]. Third, in cluster C, we observed changes in acetylation of transcriptional regulatory proteins (e.g., histones) that are suggestive of widespread transcriptional down-regulation in response to TKIs in both EGFR- and ALK-driven cancer cells. Thus, in-depth analysis of individual clusters in the CCCN revealed functional themes as well as evidence of PTM changes in response to TKIs that could mediate drug resistance.

Finally, we compared our data to previously published data from an integrated analysis of protein phosphorylation, methylation, and acetylation [15]. Despite significant differences in PTM proteomics methodology and even in the types of PTMs assayed (methylation in the previous study vs. ubiquitination in this study), we identified a core set of important, highly connected proteins in common to both networks. Several of these proteins were also identified as key proteins in our other analyses (e.g., as links between our PCN focus pathways) and many have documented roles in cancer, which confirms that our approach of using PTM clusters to examine cell signaling pathways returns actionable information. SLC25A5 and ENO1, which we found in our common core, were recently identified among six hub genes prognostic for cervical cancer [60]. SLC25A5 (ANT2) is an ADP/ATP translocase associated with several cancers [61] (https://www.proteinatlas.org/ENSG00000005022-SLC25A5/pathology). Similarity to related SLC25A4 (ANT1) suggests the possibility that SLC25A5 (ANT2) may play a role as a low-conductance pore-forming component of the mitochondrial permeability transition pore (mPTP) [62]. The expression level of ENO1 (ENOA), a glycolytic enzyme the catalyzes the conversion of 2-phosphoglycerate to phosphoenolpyruvate, is diagnostic for several cancers, including lung cancer [63–65] (https://www.uniprot.org/uniprot/P06733).

Another common core protein, PKM (pyruvate kinase M), has two isoforms (1 and 2) that convert phosphoenolpyruvate (PEP) to pyruvate in glycolysis and are implicated in the Warburg effect [66,67]. PKM is modified by phosphorylation, acetylation and ubiquitination in our data and is connected to three RTKs and four SFKs in our CFN (Fig 4B). FASN (fatty acid synthase), was also part of the common core (Fig 7). FASN is being explored as a drug target because it enables cancer cells to survive in tumors [68,69]. FASN's activity and expression has been shown to be controlled by EP300 acetylation and ubiquitination [70,71]. FASN ubiquitination on multiple sites was detected in this study and two sites were up-regulated in response to erlotinib (K70; K1993; S4 Table). Our analysis suggests the hypothesis that other proteins that also appear as links between EGF/EGFR and our focus pathways (Figs 3 and 4) could be candidates for drugs that can be used in combination therapy for cancer.

The common core also had a high concentration of negatively correlated PTMs, including many examples of acetylation and ubiquitination on the the same site (S14C Fig and S1 Table). This supports the central importance of reciprocally antagonistic signaling events in regulation of cellular signaling [15]. TKIs have been a focus for cancer therapy because they target hyperactive oncogenic kinases. A potential application of our findings is to selectively target enzymes involved in acetylation and ubiquitination, and protein interactions involving recognition of these modifications, as an additional means to modulate specific cell signaling pathways to mitigate chemotherapy resistance [15,72–74].

There are still large gaps in our knowledge about the regulatory function of many PTMs. We encountered numerous PTMs on proteins of interest (e.g., proteins in our focus pathways or proteins that were key intermediates) that were strongly affected by TKIs where little is known about the biological effect of the modification. In most cases, these PTMs have been observed in high-throughput PTM proteomics analyses but have never been the subject of focused studies. Thus, another useful outcome of this work is to highlight understudied PTMs that should be prioritized for future in-depth studies, for example, drug affected PTMs that co-cluster with PTMs known to be involved in RTK signaling (cluster A, Fig 5); or drug affected PTMs of members of the common core (S14C Fig).

In this article, we focused on interactions among EGFR signaling, glycolysis, and small molecule transport because of the strong evidence for these connections provided by our PTM data and because these pathways have an established, important role in lung cancer; however, numerous other relevant interactions can be explored. The networks can be queried as navigable data structures in multiple ways for further investigation (NDEx https://www.ndexbio.org/viewer/networks/452ac60b-4681-11ed-b7d0-0ac135e8bacf).

## Methods

### Lung cancer cell lines and drug treatments

Five tyrosine kinases (TK): EGFR, ALK fusion, ROS, ERBB2/HER2, and DDR2, represent the major known oncogenes driving NSCLC. Afatinib, dasatinib, erlotinib, and crizotinib target these five TKs and have been used clinically to treat lung cancer. To understand the crosstalk among signaling pathways mediated by major protein post-translational modifications including phosphorylation, ubiquitination, and acetylation, we selected ten NSCLC cell lines harboring driver tyrosine kinases and treated them with and without corresponding TKIs (Fig 1B). We carried out triplicate experiments on EML4-ALK-driven H3122 cells and EGFR-driven PC9 cells treated with and without TKIs, including crizotinib and erlotinib. A similar drug treatment experiment was then performed on ten different cell lines, including EGFR-driven PC9, HCC827, and H4006 cell lines, EML4-ALK-driven H3122, H2228, and STE1 cell lines, ROS1-driven HCC78 cell line, ERBB2-driven H1781 cell line, and DDR2-driven H2286 and

H366 cell lines. Afatinib (ERBB2/HER2+), Erlotinib (EGFR+), Crizotinib (ALK+, or ROS1+), and Dasatinib (DDR2+) were used as TKIs for the corresponding targets. DMSO vehicle served as the control for all TKIs. All inhibitors were obtained from Selleck Chemicals (Houston, TX). Cells were treated with TKIs at 1uM for three hours, which significantly decreased cell viability without killing cells.

## Sequential enrichment of multiple global post translational modification

Peptides modified by phosphorylation, ubiquitination, and acetylation were sequentially enriched and purified using PTMScan Phospho-Tyrosine Rabbit mAb (P-Tyr-1000) Kit (Cell Signaling Technology #8803), PTMScan Ubiquitin Remnant Motif (K-ε-GG) Kit (Cell Signaling Technology #5562), and PTMScan Acetyl-Lysine Motif (Ac-K) Kit (Cell Signaling Technology #13416). The experiment was conducted according to the kit's instructions. Briefly, for each condition, 10 x 150 mm tissue culture dishes (approximately 1–2 x 10^8 cells total) were rinsed with 5 ml PBS. The cells were lysed using 10 ml of 9M urea lysis buffer in the first dish, and then the lysate was scraped iteratively into subsequent dishes. Cell lysates were cleared by centrifuging at 20,000 x g for 15 min at room temperature, following three sonications of 15 sec each at 15 W output. The clear cell supernatants were digested with Trypsin-TPCK (Worthington, LS003744) after reduction with dithiothreitol (DTT), alkylation with iodoacetamide (IAA), and dilution with 20 mM HEPES buffer (pH 8.0). The resulting peptides were acidified and desalted using the Sep-Pak C18 Classic Cartridge (Waters). After purification, the peptides were lyophilized and resuspended in immunoaffinity purification (IAP) buffer (PTMScan, Cell Signaling Technology, Beverly, MA). The supernatant containing pure peptides was centrifuged for 5 minutes at 10,000 x g at 4˚C in a microcentrifuge, and then incubated with Phospho-Tyrosine Motif (Y*) (P-Tyr-1000) Immunoaffinity Beads. Tyrosine phosphorylated peptides and antibody complexes were separated by centrifugation at 4˚C for one minute at 1,000 x g. Supernatant was then incubated with Ubiquitin Remnant Motif (K-ε-GG) Antibody Beads. Following the isolation of ubiquitinated peptides, the acetylation peptide separation with Ac-K beads was performed in a similar fashion. After washing, all peptides were separately eluted from beads using 0.15% TFA and stored at -80˚C for further analysis.

## Protein and PTM identification and quantification using liquid chromatography tandem mass spectrometry and database searching

Nanoflow ultra-high performance liquid chromatography (RSLCnano, Dionex, Sunnyvale, CA) and a hybrid quadrupole-Orbitrap mass spectrometer (Q Exactive Plus, Thermo, San Jose, CA) were used for tandem mass spectrometry peptide sequencing. The sample was first loaded onto a precolumn (C18 PepMap100, 2 cm in length × 100 μm ID packed with C18 reversed-phase resin, 5 μm particle size, 100 Å pore size) and washed for 8 minutes with aqueous 2% acetonitrile and 0.1% formic acid. The trapped peptides were eluted on the analytical column (C18 PepMap100, 75 μm ID × 50 cm in length, 2 μm particle size, 100 Å pore size [Thermo, Sunnyvale, CA]) using a 120-minute gradient programmed as: 95% solvent A (aqueous 2% acetonitrile + 0.1% formic acid) for 8 minutes, 5% to 38.5% solvent B (aqueous 90% acetonitrile + 0.1% formic acid) for 90 minutes, 50% to 90% solvent B for 7 minutes and held at 90% for 5 minutes; which was followed by 90% to 5% solvent B for 1 minute and re-equilibration for 10 minutes. A flow rate of 300 nL/minute was used on the analytical column. NanoESI spray voltage was 1900 v and capillary temperature was 275˚C. Resolution setting for MS1 and MS/MS was 70,000 and 17500, respectively. Sixteen tandem mass spectra were collected in a data-dependent manner following each survey scan. Sixty second exclusion was used for previously sampled peptide peaks.

MaxQuant software [75,76] was employed to assign the PTM sites in the peptides and quantify PTM-modified peptides. To identify proteins and assign the PTM sites, raw data files were submitted to MaxQuant 1.5.3.30 software to search the Uni-Prot human canonical and isoform database version 04.29.2016. The sequences were reversed and added into the database to estimate false discovery rate. Three modifications were separately searched: phosphorylation on serine, threonine, and tyrosine residues; Gly-Gly modification on lysine (to detect ubiquitination); and acetylation on lysine. The false discovery rate cutoff for proteins was set at 0.05. Entries with contamination and reverse identification prefixes, such as "CON_" and "REV_", were removed from the final protein ID results. Three variant modifications setting are 1) Oxidation (M)/Acetyl (protein N-term)/phosphorylation -phosphor (STY), 2) Oxidation (M)/ Acetyl (protein N-term)/ubiquitination-glygly (K), 3) Oxidation (M)/Acetyl (protein N-term)/ acetyl(K). PTM sites maximum in each peptide was set at 3.

PTM data were used to calculate treatment:control ratios, which were limited to +/-100 to avoid statistical bias and transformed into $\log_2$ values as described previously [15]. Ratios were calculated by comparing each drug-treated sample to the average of control replicates. Ratio data and original data were both used as feature vectors for PTM clustering.

## Clustering of PTMs and construction of CCCN and CFN

Construction of the CCCN and CFN were done as described [13–15]. Briefly, pairwise-complete Euclidean distance, Spearman and hybrid Spearman-Euclidean dissimilarity (SED) was calculated using R from the combined data from the SEPTM experiments described. t-SNE embeddings were created using Rtsne using the Barnes-Hut implementation of t-SNE [25] (https://github.com/jkrijthe/Rtsne). Clusters were identified using the minimum spanning tree method from three three dimensional t-SNE embeddings. Each cluster represents the intersection of all three t-SNE embeddings. To construct a co-cluster correlation network (CCCN), an adjacency matrix was constructed by pairing co-clustered modifications to each other. Spearman correlation values were used as network edge weights. Additionally, negative correlations (<-0.5) between different PTM types on the same protein were added as CCCN edges to examine reciprocally antagonistic relationships when graphing PTM CCCN edges. This PTM CCCN was used to construct a protein CCCN by merging all co-clustered PTMs into the gene names of modified proteins. The final protein (gene) CCCN represents the sum of all PTM ratios into protein nodes and PTM correlation edge weights among all proteins (genes) whose PTMs clustered together and were detected in two or more experiments.

The resulting protein CCCN was used to create a cluster-filtered network (CFN) of known PPI interactions that were filtered by excluding all interactions save those from proteins with co-clustered modifications. The PPI dataset was composed of curated physical interactions with a focus on direct interactions as described [15] from STRING [19], GeneMANIA [17,26], BioPlex [18], Pathway Commons [16], and the kinase-substrate data from PhosphositePlus [27]. Networks were graphed in Cytoscape using RCy3 [77,78]. Additional analyses using the R programming language employed igraph [79] and bioconductor packages [80,81]

## Pathway crosstalk network (PCN) construction

The PCN was constructed using R. BioPlanet pathways were downloaded from the NCATS website (https://tripod.nih.gov/bioplanet/) as gene sets and converted into a list of pathway names whose elements are the genes in each pathway. In the PCN, each BioPlanet pathway was represented as a node. Pathways were joined by edges weighted according to the PTM Cluster Weight (pathway interaction based on co-clustering of PTMs), Gene Ontology (GO) similarity (pathway interaction based on common GO annotations), or Jaccard Similarity

(pathway interaction based on shared protein members), which are described in detail below. When only pathway interactions with non-zero PTM Cluster Weight are included, the PCN contained 1488/1658 BioPlanet pathways and 645,709 edges out of 1,106,328 possible edges (density = 0.5836506) (S7 Table). The version of the PCN that only includes interactions with non-zero PTM Cluster Weight and zero Jaccard Similarity also contained 1488 pathways, but only 456,985 edges (density = 0.4130647).

## PTM Cluster Weight

We hypothesized that PTM clusters from the CCCN may be used to identify relationships among cell signaling and other pathways that are active in lung cancer cell lines. For each pair of BioPlanet pathways, we calculated PTM Cluster Weight based on the extent to which the proteins in each pathway have PTMs that co-cluster in the CCCN. PTM Cluster Weight was adjusted to reduce the influence of large clusters and of proteins that were found in many pathways.

To calculate PTM Cluster Weight, we first considered each cluster (i) and for each pathway (j) calculated a score (Cluster Pathway Evidence (i,j)) that captures the representation of pathway (j) in cluster (i):

$$CPE\left(cl_i, pw_j\right) = \sum_{pro_k \in pw_j} \frac{|\{PTM_x | PTM_x \in pro_k \cap PTM_x \in cl_i\}|}{|\{pw_y | pro_k \in pw_y\}| \times |cl_i|} \tag{1}$$

CPE = Cluster Pathway Evidence
$pro_{(k)}$ = protein (k)
$PTM_{(x)}$ = PTMs on protein (x)
$pw_j$ = pathway (j)
$cl_i$ = cluster (i)
denominator = total number of Pathways in which protein (k) is found * size of cluster (i)
In other words, Cluster Pathway Evidence (i,j) = $\Sigma_k$(# of PTMs on protein (k) in pathway (j) found in cluster (i) / [total number of Pathways in which protein (k) is found * size of cluster (i)].

Weights for PTMs with ambiguous protein assignments (where peptide sequences are identical in several proteins) were divided by the number of ambiguous proteins, so that ambiguous PTMs were weighted proportionally to the number of possible matches.

The result was a matrix with pathway names in columns and individual clusters as rows. This matrix was interpreted as a bipartite graph where PTM clusters form one projection and BioPlanet pathways form the second projection. An edge table was constructed consisting of pathway pairs that have non-zero Cluster Pathway Evidence from the same cluster(s). For each pathway pair, Cluster Pathway Evidences for both of the pathways were summed for all clusters where both pathways were represented.

To illustrate how this method works, consider the two BioPlanet pathways "Transmembrane transport of small molecules" and "EGF/EGFR signaling pathway." There were 68 clusters that contained PTMs on one or more proteins from the "Transmembrane transport of small molecules pathway" *and* on one or more proteins from the "EGF/EGFR Signaling Pathway". The individual Cluster Pathway Evidences for the two pathways ranged from 0.0278 to 0.122 for these 68 clusters and summed to 1.659, which is strong evidence for interactions between these two pathways.

The raw weights were then normalized to fall into the range 0–1 to produce the final PTM Cluster Weights. Sub-networks of the PCN that were selected by thresholding the various edge weights were graphed in Cytoscape using RCy3 [77,78].

## Gene Ontology (GO) similarity

We defined the similarity of a pathway pair in three steps as the average similarity of their genes as measured by their GO Biological Process (GOBP) annotations. First, we used the Gene Ontology to define a vector of a gene's GOBP annotations (*i.e.*, GOBP terms associated with the gene). Next, we calculated the similarity of the two genes using the Otsuka–Ochiai coefficient (which is equivalent to the cosine similarity):

$$GeneSim(gene_A, gene_B) = \frac{|GOBP_A \cap GOBP_B|}{\sqrt{|GOBP_A| \times |GOBP_B|}} \tag{2}$$

where,

 $GOBP_A$ = set of GOBP annotations for gene A

 $GOBP_B$ = set of GOBP annotations for gene B

We computed gene similarity for every gene pair ($gene_A$, $gene_B$) from every Bioplanet pathway pair ($Pathway_i$, $Pathway_j$). Note that each pathway had a different number of genes, and large pathways may create too many low gene similarity values. We carefully tailored our similarity computations to avoid bias against large pathways in the following way. Once we computed all gene similarities among two pathways, we considered the top-N similar gene pairs to compute the pathway similarity:

$$GOSim\left(Pathway_i, Pathway_j\right) = \frac{1}{N} \sum_{n=1}^{N} GeneSim\left(gene_{n,i}, gene_{n,j}\right) \tag{3}$$

where $gene_{ni}$ *and* $gene_{nj}$ are the nth highest similarity gene pair from $Pathway_i$ and $Pathway_j$. Note that we do not use shared genes in GOSim computations (*i.e.*, $gene_{ni} \neq gene_{nj}$).

A crucial step in GOSim is to specify the value of *N*. A large *N* value penalizes large pathways, whereas a small *N* value ignores useful gene similarity information beyond the *Nth* pair. We experimented with $N \in (10, 30, 50, 100)$. With this approach, we achieved the best results with *N = 30* (see S1 Text for more details).

## Jaccard similarity

Finally, for each pathway pair, we calculated the Jaccard similarity as a measure of the degree to which the pathways were similar due to sharing genes in common:

The Jaccard similarity is the ratio of the number of elements in the intersection of the two sets to the number of elements in the union of the sets.

$$J\left(P_i, P_j\right) = \frac{|P_i \cap P_j|}{|P_i \cup P_j|} \tag{4}$$

## Cluster enrichment analysis

Clusters in the CCCN that are significantly enriched for drug-affected PTMs were identified for the following groups of samples: (i) h3122.criz: H3122 cells treated with crizotinb (5 samples); (ii) hcc78.criz: HCC78 cells treated with crizotinib (2 samples); (iii) pc9.erl: PC9 cells treated with erlotinib (5 samples); all.criz: H3122 (5 samples), HCC78 (2 samples), H2228 (1 sample), and STE1 (1 sample) cells treated with crizotinib; (iv) all.erl: PC9 (5 samples), HCC4006 (1 sample), and HCC827 (1 sample) cells treated with erlotinib; (v) all.dasat: H2286 (1 sample) and H366 (1 sample) cells treated with dasatinib; (vi) all.drug: all samples from all. criz, all.erl, and all.dasat as well as H1781 cells (1 sample) treated with afatinib. The drug-treated vs. control ratio was calculated for each PTM in each sample, and, for each sample

group, PTMs with at least two observations and a median ratio of at least 2.25-fold up or down (ratio > 1.17 or < -1.17 on a $\log_2$ scale) were considered drug-affected (2.25-fold change was chosen to ensure robust statistical power). Drug-affected PTMs were mapped to CCCN clusters. For each sample group, for each cluster, a 2x2 contingency matrix was constructed with the following values: the number of drug-affected sites in the cluster, the number of drug-affected sites not in the cluster (*i.e.*, found in other clusters), the number of remaining (not drug-affected) sites in the cluster, and the number of remaining (not drug-affected) sites not in the cluster, and a one-sided Fisher Test was performed in R to identify clusters with more drug-affected sites than expected by chance. p-values were corrected for multiple testing using the Benjamini-Hochberg method. To limit the number of tests, only clusters with at least three drug-affected sites were included in the analysis. Clusters with corrected p-values < 0.05 were considered significantly enriched (S5 Table).

## Cluster heatmaps

Heatmaps were constructed using the R function heatmap.2 (gplots package). They display sample groups (defined in the Cluster Enrichment Analysis section) as columns and PTM sites from clusters of interest as rows. Cells were colored to indicate either: (i) the median of the $\log_2$ ratio of each PTM site in drug-treated vs. control cells or (ii) sites that met the threshold to be selected for the enrichment analysis; sites with median $\log_2$ ratios less than -1.17 (2.25-fold down) in each sample group are shown in blue, and sites with median $\log_2$ ratios greater than 1.17 (2.25-fold up) are shown in yellow (black indicates sites that did not meet the threshold threshold). Sites that met the down threshold were assigned a value of -1, sites that met the up threshold were assigned a value of +1 and sites (rows) in the heatmap were ordered according to the sum of these values across all sample groups.

## K-core decomposition

A k-core $G^k$ of a graph G is the subgraph of G obtained by iteratively deleting all vertices (and edges connected to it) with a degree less than k [43]. In other words, $G^k$ is the largest subgraph of G where all the vertices have a degree of at least k.

k-core decomposition ranks vertices of a graph according to their connections and creates groups of increasingly more 'important' or 'central' vertices. Such decomposition allows a clarified view on vertices and facilitates researching the role of certain vertex groups in isolation. We applied k-core decomposition to the CFN generated in this study and the CST CFN to identify the compositional differences between the two graphs in terms of the highest cores. The graphs are not simple; 93 edges in our CFN and 47 edges in CST CFN connect two vertices V2 and V1, where an edge from V1 to V2 already exists. In the rest of our analysis, we use the simplified versions of the two networks where we omit self loop edges of vertices and duplicate edges of vertex pairs (see S1 Text).

## Supporting information

**S1 Fig. PTM changes in response to different TKIs.** (A) Overlap of individual PTMs that were more than 2.25-fold increased or decreased by different drugs, grouped by PTM type. (B) Density plots showing changes greater than 2.25 fold for each PTM and drug. Decreased PTM amounts in response to drug are highlighted in blue; increased amounts in yellow. Changes below the 2.25-fold threshold are omitted to highlight significant changes. (C) Correlation density between different PTMs on the same proteins for phosphorylation and acetylation; phosphorylation and ubiquitination; and acetylation and ubiquitination. Negative correlations selected for display as edges are highlighted in blue; positive correlations are highlighted in

yellow.
(PDF)

**S2 Fig. Construction of the co-cluster correlation network (CCCN).** (A) Example three dimensional t-SNE embedding of PTM data using Spearman-Euclidean dissimilarity (SED) from PTM data. Co-clustered PTMs that are close to one another and share the same color. (B) CCCN of all PTMs. Yellow edges are positive correlation between co-clustered PTMs; blue edges are negative correlations, negative correlations < -0.5 among different PTMs on the same protein are included even if these PTMs do not co-cluster. (C) Node and edge key. Protein families are indicated by node shape and border color (left). Edges that represent interactions between proteins are colored according to interaction type (right); PTM correlations edges are at the bottom of the list. Edges that connect proteins to their PTMs in combined CFN/CCCN graphs are black.
(PDF)

**S3 Fig. TKI effects on RTK and SFK PTMs.** (A) Heatmap showing PTM $\log_2$ fold changes (key below A) on all RTK and SFK PTMs, sorted by hierarchical clustering (dendrogram at left). (B-D) CCCN interactions among RTK PTMs (left) and SFK PTMs (right). Node size and color represents $\log_2$ fold change (bar under B): blue is down-regulated; yellow up-regulated, for cells treated with erlotinib (B), crizotinib (C), and dasatinib (D). Node border and shape and edge colors are defined in S2C Fig. Edges connecting proteins to their PTMs were colored light grey for clarity.
(PDF)

**S4 Fig. Cluster-filtered network (CFN) and combined CFN/CCCN.** (A) CFN of PPIs filtered by co-clustered PTMs. (B) Combined CFN and CCCN. Node shape and outline color, and edge colors, are defined in S2C Fig.
(PDF)

**S5 Fig. Comparison of shortest paths networks from drug targets to drug-affected sites.** Heatmap depicting, for each pair of samples, the Jaccard similarity (intersection divided by union) of the sets of genes comprising the shortest paths in the CFN connecting genes with significantly changed PTMs to the mutant driver protein/drug target. PTMs with a fold-change of at least 2.25 in drug-treated vs. control cells were defined as significantly changed. Drug/ driver protein pairs were as follows: Crizotinib-ALK, Erlotinib-EGFR, Afatinib-ERBB2/HER2.
(PDF)

**S6 Fig. Comparison of PTM cluster weight with Jaccard similarity and Gene Ontology (GO) Biological Process similarity for the BioPlanet Pathway Crosstalk Network.** Pathway-pathway interactions were defined based on PTM clustering, BioPlanet Jaccard similarity, or GO similarity (see Methods). (A) PTM cluster weight was poorly correlated with the BioPlanet Jaccard similarity ($R^2$ = 0.02037). (B) GO similarity (not including genes in common between pathways) was moderately correlated with BioPlanet Jaccard similarity ($R^2$ = 0.2208). (C) PTM cluster weight vs. GO similarity ($R^2$ = 0.1336). Although 13% of pathway pair edges had zero GO similarity (points at 0 on the x-axis of the graph), these edges represent only a small fraction (0.34%) of the total PTM cluster weight in the network. Without these edges of zero GO similarity weight, the the $R^2$ correlation between PTM cluster weight and GO similarity was 0.1508. (D) Same as C but excluding pathway pairs that have one or more genes in common (*i. e.*, only interactions with Jaccard similarity = 0 were plotted; $R^2$ = 0.06078).
(PDF)

**S7 Fig. Pathway Crosstalk Network (PCN) strongest cluster-based associations.** PCN filtered to show pathway pairs with strong PTM cluster weight ($> 0.05$, purple edges) and no genes in common (Jaccard similarity = 0). 997 out of 455988 pathway-pathway edges are shown. The pathways EGF/EGFR signaling pathway, Glycolysis and gluconeogenesis, and Transmembrane transport of small molecules are highlighted in yellow.
(PDF)

**S8 Fig. Response of EGFR signaling and transporter PTMs to TKIs.** PTM clusters containing transporters and EGFR. (A, B, C) ABC transporter PTMs and CFN pathways connecting ABC transporters to the TKI targets EGFR, ALK, and MET. (D, E, F) PTM clusters that contain EGF/EGFR and transmembrane transporter pathway PTMs. Node size and color (scale bar in A) indicates response to erlotinib (A, D), crizotinib (B, E), and dasatinib (C, F). Node border and shape and edge colors are defined in S2C Fig.
(PDF)

**S9 Fig. Crizotinib-affected PTMs linked to EGFR signaling.** Direct interactions in the CFN between members of the EGF/EGFR signaling pathway and the Transmembrane transport of small molecules pathway (A) and Glycolysis and gluconeogenesis pathway (B). PTMs that were significantly changed ($>2.25$-fold) in response to crizotinib in H3122 cells are shown. Node border and shape and edge colors are defined in S2C Fig.
(PDF)

**S10 Fig. EGFR glycolysis networks.** (A) Combined CFN/CCCN showing composite shortest paths from the BioPlanet pathways EGF/EGFR signaling pathway (top) and Glycolysis and gluconeogenesis (bottom), graphed as in Fig 3. (B) Same as A but showing CFN edges only. (C) "Mutual friends" (center row) defined as proteins that connect to at least one member of both pathways in the CFN. Node size and color represents $\log_2$ fold change (bar in B) for all TKIs (A, B) and erlotinib (C).
(PDF)

**S11 Fig. Drug-affected PTMs in cluster A.** (A) CFN for proteins with PTM sites in Cluster A. Node size and color indicates the $\log_2$ ratio of PTM abundance (averaged over all PTMs for a protein) in dasatinib treated vs. untreated cells. Node border and shape and edge colors are defined in S2C Fig. (B) Heatmaps showing TKI responses of PTM sties in cluster A. Left panel: Sites in cluster A with drug-treated to control ratios that were at least 2.25-fold higher in drug-treated cells or at least 2.25-fold lower in drug-treated cells are indicated in yellow and blue, respectively. Black indicates sites that were below threshold. Right panel: the drug-treated to control ratio (median of samples in each group) for each PTM site is shown. Sample groups are as in Fig 5.
(PDF)

**S12 Fig. Drug-affected PTMs in cluster B.** Heatmaps showing TKI responses of PTMs in cluster B. Left panel: Sites in cluster B with drug-treated to control ratios that were at least 2.25-fold higher in drug-treated cells or at least 2.25-fold lower in drug-treated cells are indicated in yellow and blue, respectively. Black indicates sites that were below threshold. Right panel: the drug-treated to control ratio (median of samples in each group) for each PTM site is shown (see key at bottom; black represents missing data). Sample groups are as in Fig 5.
(PDF)

**S13 Fig. Drug-affected PTMs involved in transcription regulation.** CFN for proteins with PTM sites in Cluster C. Note that there are fewer nodes than there are PTM proteins in the cluster because some of the cluster proteins did not have any physical interactions with each

other in the PPI databases used for CFN construction. Node size and color indicates the $\log_2$ ratio of PTM abundance (averaged over all PTMs for a protein) in PC9 erlotinib treated vs. control cells. Node border and shape and edge colors are defined in S2C Fig.
(PDF)

**S14 Fig. Comparison of networks (CFNs) from different lung cancer PTM data sets.** (A) Shared core matrix. The CST-CFN from a previous study [15] is plotted on the x-axis; the CFN from this study on the y-axis. The numbers indicate the number of proteins in common in each pair of cores. Cells are colored yellow, orange and red by the number of common proteins. Note that the CST-CFN lacks an 8th core, and the 9th and 10th cores represent a clique of interconnected ribosomal proteins. (B) Number of proteins in each core found in the CFN from this study (left) and the CST-CFN (right). (C) PTMs detected in this study from 43 proteins that appear in the high cores of both CFNs (Fig 7) graphed as a CCCN using data from this study. Node size and color reflects PTM changes in cells treated with all TKIs.
(PDF)

**S1 Text. This document provides more details on shortest path subnetworks, validation of PTM cluster weight for pathway similarity, enrichment of CCCN clusters for drug-affected PTMs, k-core analysis, and GOsim score for pathway similarity.**
(DOCX)

**S1 Table. Quantitation of unique PTMs (acetylation, phosphorylation, and ubiquitination) identified using SEPTM proteomics.**
(XLSX)

**S2 Table. PTMs found in each cluster in the CCCN.**
(TXT)

**S3 Table. Pathway-pathway interactions from the PCN involving the EGF/EGFR signaling pathway.**
(TXT)

**S4 Table. Drug-affected sites (median absolute log2 fold-change at least 1.17) in drug-treated vs. control cells) in each drug/cell-line sample group.**
(XLSX)

**S5 Table. CCCN clusters significantly enriched (corrected p-value < 0.05) in drug-affected sites from each drug/cell-line sample group.**
(XLSX)

**S6 Table. Annotation of PTM sites in selected CCCN clusters.**
(XLSX)

**S7 Table. Pathway-pathway interactions in the PCN.**
(TXT)

## Author Contributions

**Conceptualization:** Karen E. Ross, Guolin Zhang, Cuneyt Akcora, John Koomen, Eric B. Haura, Mark Grimes.

**Data curation:** Karen E. Ross, Guolin Zhang, Bin Fang, John Koomen, Mark Grimes.

**Formal analysis:** Karen E. Ross, Cuneyt Akcora, Yu Lin, Mark Grimes.

**Funding acquisition:** Eric B. Haura.

**Investigation:** Karen E. Ross, Guolin Zhang, Cuneyt Akcora, Mark Grimes.

**Methodology:** Karen E. Ross, Guolin Zhang, Cuneyt Akcora, Yu Lin, Bin Fang, Mark Grimes.

**Project administration:** Guolin Zhang, Eric B. Haura, Mark Grimes.

**Software:** Karen E. Ross, Cuneyt Akcora, Mark Grimes.

**Supervision:** Karen E. Ross, John Koomen, Eric B. Haura, Mark Grimes.

**Validation:** Karen E. Ross, Guolin Zhang, Yu Lin, Bin Fang, John Koomen, Mark Grimes.

**Visualization:** Karen E. Ross, Guolin Zhang, Yu Lin, Mark Grimes.

**Writing – original draft:** Karen E. Ross, Guolin Zhang, Mark Grimes.

**Writing – review & editing:** Karen E. Ross, Guolin Zhang, Mark Grimes.

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
