## [Decision Letter · Decision Letter 0]

18 Dec 2022

Dear Dr. Grimes,

Thank you very much for submitting your manuscript "Network models of protein phosphorylation, acetylation, and ubiquitination connect metabolic and cell signaling pathways in lung cancer" for consideration at PLOS Computational Biology.

As with all papers reviewed by the journal, your manuscript was reviewed by members of the editorial board and by several independent reviewers. In light of the reviews (below this email), we would like to invite the resubmission of a significantly-revised version that takes into account the reviewers' comments.

We cannot make any decision about publication until we have seen the revised manuscript and your response to the reviewers' comments. Your revised manuscript is also likely to be sent to reviewers for further evaluation.

Sincerely,

Feixiong Cheng, Ph.D.

Guest Editor

PLOS Computational Biology

Jian Ma

Section Editor

PLOS Computational Biology

Reviewer's Responses to Questions

**Comments to the Authors:**

Reviewer #1: Systematic analysis of Protein post-translational modifications (PTMs) data in lung cancer can try to model cell signaling pathways. Uncovering crosstalk among signaling pathways employing different PTMs offer potential drug targets and candidates for synergistic attack through combination drug therapy. To model lung cancer signaling at the protein level, the authors applied SEPTM proteomics to identify PTMs, and constructed the Co-Cluster Correlation Network (CCCN) and Cluster Filtered Network (CFN). They then constructed a Pathway Crosstalk Network (PCN) to identify relationships among known biological pathways in human cells. Furthermore, they compared their result with previously published data, and supported their analysis result. Generally, this study have done a beautiful and integrative work. The results suggest unappreciated connections between RTK signal transduction and oncogenic metabolic reprogramming in lung cancer. However, several minor problems should be acknowledged in this study.

1)The main text is very long in this manuscript, which makes the readers hard to follow. I suggest some of the main text should be removed as supplementary method ;

2)The resolution of some figures is low. They should be replaced with high resolution ones;

3)The authors compared networks (CFNs) from different lung cancer PTM data sets, and reveals a common core of PPIs. The more potential application of this finding in future should be discussed.

Reviewer #2: The manuscript, entitled “Network models of protein phosphorylation, acetylation, and ubiquitination connect metabolic and cell signaling pathways in lung cancer”, has studied signaling pathways disturbed by different tyrosine kinase inhibitors (TKIs) in ten lung cancer cell lines. With the help of Network models in the manuscript, the author showed the changes of affected cell lines in cell signaling pathways activated by oncogenes, transmembrane transport of small molecules, and glycolysis and gluconeogenesis. In my opinion, this work is interesting and helpful for Multi-drug Therapy for Cancer in the future. Therefore, I support the publication of this manuscript after the following question are followed:

1. In Introduction, the author introduced the concepts and importance of Protein post-translational modifications、lung cancer and their hypothesis, making it easier to understand the study’s background and targets. However, the motivation and innovation of this project is not very clearly in this part. I suggest that the motivation, innovation as well as the main contribution of this project should be summarized in the Introduction section.

2. The authors try to use Figure 1 to illustrate the main strategy of this project. However, I don’t think this figure is suitable to show the pipeline of the methods (at least it not very obviously). I suggest the authors to reorganize the Figure 1, which would make the work more understandable.

3. After the introduction the network model, the results focus on the analysis of EGFR. And it seems that the network models just verify the importance of the EGFR. How we can learn from the network model is not very clear. And more statistic analysis of the network models should be provided.

4. There are many methods to identify the core of network, such as the mining of the dense subgraph. The authors used the k-core decomposition. And what about the result of the other methods?

5. In page 8, the authors stated that “chose one target for each drug …”. Why only one target was selected? How to choose the target for more than one targets of the drug?

6. The statement “Cluster Pathway Evidence (i,j) =…” in page 32. I suggest the authors can use the mathematical formula to describe the calculation. As well as the GeneSim, Jaccard Similarity.

7. In the definition of the GoSim, the authors state that they achieved the best results with N=30. What’s the meaning of the best results? Are there any other parameters used in the network models? And what’s the influence?

Reviewer #3: Review has been uploaded as an attachment.

**Have the authors made all data and (if applicable) computational code underlying the findings in their manuscript fully available?**

Reviewer #1: Yes

Reviewer #2: None

Reviewer #3: Yes

PLOS authors have the option to publish the peer review history of their article (what does this mean?). If published, this will include your full peer review and any attached files.

Reviewer #1: No

Reviewer #2: No

Reviewer #3: No
---

## [Decision Letter · Decision Letter 1]

11 Mar 2023

Dear Dr. Grimes,

We are pleased to inform you that your manuscript 'Network models of protein phosphorylation, acetylation, and ubiquitination connect metabolic and cell signaling pathways in lung cancer' has been provisionally accepted for publication in PLOS Computational Biology.

Best regards,

Feixiong Cheng, Ph.D.

Guest Editor

PLOS Computational Biology

Jian Ma

Section Editor

PLOS Computational Biology

Reviewer's Responses to Questions

**Comments to the Authors:**

Reviewer #1: It can be accepted for publication now

Reviewer #2: All my comments are well addressed, and I recommend its publication on PLoS Computational Biology.

Reviewer #3: I think that authors have addressed the comments I made last time.

**Have the authors made all data and (if applicable) computational code underlying the findings in their manuscript fully available?**

Reviewer #1: Yes

Reviewer #2: Yes

Reviewer #3: Yes

PLOS authors have the option to publish the peer review history of their article (what does this mean?). If published, this will include your full peer review and any attached files.

Reviewer #1: No

Reviewer #2: No

Reviewer #3: No

---

## [Editor Report · Acceptance letter]

22 Mar 2023

PCOMPBIOL-D-22-01566R1 

Network models of protein phosphorylation, acetylation, and ubiquitination connect metabolic and cell signaling pathways in lung cancer

Dear Dr Grimes,

I am pleased to inform you that your manuscript has been formally accepted for publication in PLOS Computational Biology. Your manuscript is now with our production department and you will be notified of the publication date in due course.

With kind regards,

Dorothy Lannert
